# An Optimal Structured Zeroth-order Algorithm for Non-smooth Optimization

**Marco Rando**[*]    Cesare Molinari[†]    Silvia Villa[†]    Lorenzo Rosasco[*‡]

## Abstract

Finite-difference methods are a class of algorithms designed to solve black-box optimization problems by approximating a gradient of the target function on a set of directions. In black-box optimization, the non-smooth setting is particularly relevant since, in practice, differentiability and smoothness assumptions cannot be verified. To cope with nonsmoothness, several authors use a smooth approximation of the target function and show that finite difference methods approximate its gradient. Recently, it has been proved that imposing a structure in the directions allows improving performance. However, only the smooth setting was considered. To close this gap, we introduce and analyze O-ZD, the first structured finite-difference algorithm for non-smooth black-box optimization. Our method exploits a smooth approximation of the target function and we prove that it approximates its gradient on a subset of random *orthogonal* directions. We analyze the convergence of O-ZD under different assumptions. For non-smooth convex functions, we obtain the optimal complexity. In the non-smooth non-convex setting, we characterize the number of iterations needed to bound the expected norm of the smoothed gradient. For smooth functions, our analysis recovers existing results for structured zeroth-order methods for the convex case and extends them to the non-convex setting. We conclude with numerical simulations where assumptions are satisfied, observing that our algorithm has very good practical performances.

## 1 Introduction

Black-box optimization problems are a class of problems for which only values of the target function are available and no first-order information is provided. Typically, these problems arise when the evaluation of the objective function is based on a simulation and no analytical form of the gradient is accessible or its explicit calculation is too expensive [14, 45, 36, 35].

To face these problems, different methods that do not require first-order information have been proposed - see for instance [39, 47, 19, 26, 20, 30, 33] and references therein. These techniques are called derivative-free methods and a wide class of these is the class of finite-difference algorithms [31, 39, 17]. These iterative procedures mimic first-order optimization strategies by replacing the gradient of the objective function with an approximation built through finite differences in random directions.

Two types of finite-difference methods can be identified: unstructured and structured ones, depending on the way in which the random directions are generated. In the former, directions are sampled i.i.d. from some distribution [39, 17, 10, 46] while in the latter, directions have to satisfy some structural constraints, e.g. orthogonality [32, 42]. Several authors [32, 42, 5, 12] theoretically and empirically

---

[*]MaLGa-DIBRIS, University of Genova, IT (`marco.rando@edu.unige.it`, `lorenzo.rosasco@unige.it`).

[†]MaLGa - DIMA, University of Genova, Italy (`molinari@dima.unige.it`, `silvia.villa@unige.it`).

[‡]Istituto Italiano di Tecnologia, Genova, Italy and CBMM - MIT, Cambridge, MA, USA

37th Conference on Neural Information Processing Systems (NeurIPS 2023).

observed that imposing orthogonality among the directions provides better performance than using unstructured directions. Intuitively, imposing orthogonality allows us to avoid cases in which the gradient approximation is built using similar or redundant directions.

Notably, methods based on structured finite differences have been analyzed only for smooth and convex (or specific non-convex) target functions. This represents a strong limitation, since smoothness and convexity are in practice hardly verifiable, due to the nature of black-box optimization problems. The aim of this paper is the analysis of structured finite difference algorithms dropping smoothness and convexity assumptions.

For unstructured finite-difference methods, a common way to face nonsmoothness consists in introducing a smooth approximation of the target function, also known as "smoothing" [6, 16] and using it as a surrogate of the target. Although the gradient of the smoothing is not computable, different authors showed that for certain types of smoothing the unstructured finite-difference approximation provides an unbiased estimation of such a gradient - see, for example, [39, 46, 18, 22, 21, 24]. This key observation allows to prove that unstructured finite-difference methods approximate a solution in different nonsmooth settings [17, 39, 46, 18, 22, 34].

For structured finite-difference methods, the analysis in the nonsmooth setting is not available. A key step which is missing is the proof of the fact that the surrogate of the gradient built with structured directions is an estimation of the gradient of a suitable smoothing.

In this paper, we close the gap, and introduce O-ZD, a structured finite-difference algorithm in the non-smooth setting. The algorithm builds an approximation of the gradient of a smoothed target using a set of $\ell \leq d$ orthogonal directions. We analyze the procedure proving that our finite-difference surrogate is an unbiased estimation of the gradient of a smoothing. We provide convergence rates for non-smooth convex functions with optimal dependence on the dimension [17] and rates for the non-smooth non-convex setting (for which lower bounds are not known [34]). Moreover, for non-smooth convex functions we provide the first proof of convergence of the iterates for structured finite differences in this setting. For smooth convex functions, we recover the standard results for structured zeroth-order methods [32, 42]. We conclude with numerical illustrations. To the best of our knowledge, this is the first work on nonsmooth finite-difference method with structured directions.

The paper is organized as follows. In Section 2, we introduce the problem and the algorithm. In Section 3, we state and discuss the main results. In Section 4 we provide some experiments and in 5 some final remarks.

## 2 Problem Setting & Algorithm

Given a function $f : \mathbb{R}^d \to \mathbb{R}$ and assuming that $f$ has at least a minimizer in $\mathbb{R}^d$, we consider the problem to find

$$x^* \in \arg\min_{x \in \mathbb{R}^d} f(x). \tag{1}$$

In particular, we consider the non-smooth setting where $f$ might be non-differentiable. To solve problem (1), we propose a zeroth-order algorithm, namely an iterative procedure that uses only function values. At every iteration $k \in \mathbb{N}$, a first-order information of $f$ is approximated with finite-differences using a set of $\ell \leq d$ random orthogonal directions $(p_k^{(i)})_{i=1}^{\ell}$. Such orthogonal directions are represented as rotations (and reflections) of the first $\ell$ vectors of the canonical basis $(e_i)_{i=1}^{\ell}$. We set $p_k^{(i)} = G_k e_i$, where $G_k$ belongs to the orthogonal group defined as

$$O(d) := \{G \in \mathbb{R}^{d \times d} \mid \det G \neq 0 \land G^{-1} = G^{\mathsf{T}}\}.$$

Methods for generating orthogonal matrices are discussed in Appendix D. Given $G \in O(d), 0 < \ell \leq d$ and $h > 0$, we consider the following central finite-difference surrogate of the gradient information

$$g_{(G,h)}(x) = \frac{d}{\ell} \sum_{i=1}^{\ell} \frac{f(x + hGe_i) - f(x - hGe_i)}{2h} Ge_i. \tag{2}$$

Then, we introduce the following algorithm.

**Algorithm 1** O-ZD: Orthogonal Zeroth-order Descent

---

**Input:** $x_0 \in \mathbb{R}^d$, $(\alpha_k)_{k \in \mathbb{N}} \subset \mathbb{R}_+$, $(h_k)_{k \in \mathbb{N}} \subset \mathbb{R}_+$, $\ell \in \mathbb{N}$ s.t. $1 \leq \ell \leq d$
**for** $k = 1, \cdots$ **do**
    sample $G_k$ i.i.d. from $O(d)$
    $x_{k+1} = x_k - \alpha_k g_{(G_k, h_k)}(x_k)$
**end for**

---

Starting from an initial guess $x_0 \in \mathbb{R}^d$, at every iteration $k \in \mathbb{N}$, the algorithm samples an orthogonal matrix $G_k$ i.i.d. from the orthogonal group $O(d)$ and computes a surrogate $g_{(G_k, h_k)}$ of the gradient at the current iterate $x_k$. Then, it computes $x_{k+1}$ by moving in the opposite direction of $g_{(G_k, h_k)}(x_k)$. This approach belongs to the class of *structured* finite-difference methods, where a bunch of orthogonal directions is used to approximate a gradient of the target function [32, 42]. Such algorithms have been proposed and analyzed only for smooth functions. To cope with this limitation, and extend the analysis to the nonsmooth setting, we exploit a smoothing technique. For a fixed a probability measure $\rho$ on $\mathbb{R}^d$ and a positive parameter $h > 0$ called *smoothing parameter*, we define the following *smooth* surrogate of $f$

$$f_{h,\rho}(x) := \int f(x + hu) \, d\rho(u). \tag{3}$$

As shown in [6], $f_{h,\rho}$ is differentiable even when $f$ is not. In the literature, different authors have used this strategy to face non-smooth zeroth-order optimization with random finite-difference methods [18, 39, 49, 22] fixing specific smoothing distribution, but no one applied and analyze it for structured methods.

In this work, $\rho$ is the uniform distribution over the $\ell_2$ unit ball $\mathbb{B}^d$, defining the smooth surrogate

$$f_h(x) = \frac{1}{\text{vol}(\mathbb{B}^d)} \int_{\mathbb{B}^d} f(x + hu) \, du, \tag{4}$$

where $\text{vol}(\mathbb{B}^d)$ denotes the volume of $\mathbb{B}^d$. One of our main contributions is the following Lemma which shows that the surrogate proposed in (2) is an unbiased estimator of the gradient of the smoothing in (4).

**Lemma 1** (Smoothing Lemma). *Given a probability space $(\Omega, \mathcal{F}, \mathbb{P})$, let $G : \Omega \to O(d)$ be a random variable where $O(d)$ is the orthogonal group endowed with the Borel $\sigma$-algebra. Assume that the probability distribution of $G$ is the (normalized) Haar measure. Let $h > 0$ and let $g$ be the gradient surrogate defined in eq.* (2). *Then,*

$$(\forall x \in \mathbb{R}^d) \qquad \mathbb{E}_G[g_{(G,h)}(x)] = \nabla f_h(x),$$

*where $f_h$ is the smooth approximation of the target function $f$ defined in eq.* (4).

The proof of Lemma 1 is provided in Appendix A.1.

**Remark 1.** *Note that Lemma 1 holds also using $g^F_{(G,h)}$ or $g^S_{(G,h)}$ defined as*

$$g^F_{(G,h)}(x) := \frac{d}{\ell} \sum_{i=1}^{\ell} \frac{f(x + hGe_i) - f(x)}{h} Ge_i \qquad and \qquad g^S_{(G,h)}(x) := \frac{d}{\ell} \sum_{i=1}^{\ell} \frac{f(x + hGe_i)}{h} Ge_i,$$

*since $\mathbb{E}_G[g_{(G,h)}(x)] = \mathbb{E}_G[g^F_{(G,h)}(x)] = \mathbb{E}_G[g^S_{(G,h)}(x)]$. Despite these two estimators being computationally cheaper than the proposed one, we use central finite differences since they allow us to derive a better bound for $\mathbb{E}_G[\|g_{(G,h)}(x)\|^2]$ as observed in [46] for the case $\ell = 1$.*

Thanks to Lemma 1, we can interpret each step of Algorithm 1 as a Stochastic Gradient Descent (SGD) on the smoothed function $f_{h_k}$. But the analysis of the proposed algorithm does not follow from the SGD one for two reasons. First, the smoothing parameter $h_k$ changes along the iterations; second (and more importantly), the set of minimizers of $f_h$ and $f$ are different in general. However, we will take advantage of the fact that $f_h$ can be seen as an approximation of $f$. The relationship between $f$ and its approximation $f_h$ depends on the properties of $f$ - see Proposition 1 in Appendix A.

Our main contributions are the theoretical and numerical analysis of Algorithm 1 under different choices of the free parameters $\alpha_k, h_k$, namely the stepsize and the smoothing parameter. To the best of our knowledge, Algorithm 1 is the first structured zeroth-order method for non-smooth optimization.

## 2.1 Related Work

In practice, the advantage of the use of structured directions in finite-difference methods has been observed in several applications [12] and motivated their study. In [5], the authors theoretically and empirically observed that structured directions provide a better approximation of the gradient with respect to unstructured (Gaussian and spherical) ones. Next, we review the most related works, showing the differences with our algorithm.

**Unstructured Finite-differences.** Most of existing works focused on the theoretical analysis of methods using a single direction to approximate the gradient - see e.g. [39, 46, 22, 45, 18, 24, 34, 17, 10]. The main results existing so far analyze the convergence of the function values in expectation. They provide convergence rates in terms of the number of function evaluations and explicitly characterize the dependence on the dimension of the ambient space.

**Smooth setting:** In [39, 17], a finite difference algorithm with a single direction is analyzed. Rates on the expected function values are shown. In [39] the dependence on the dimension in the rates is not optimal. In [17], both single and multiple direction cases are analyzed and lower bounds are derived. However, only the convex setting is considered. In [24], both convex and non-convex settings are analyzed. They obtain optimal dependence on dimension in the complexity for convex functions. However, only the single-direction case is considered and only the smooth setting is analyzed. Comparing the result with our rates, Algorithm 1 achieves the same dependence on the dimension in the complexity taking $\ell$ as a fraction of $d$. Note that, by parallelizing the computation of the function evaluations, we can get a better result.

**Non-smooth setting:** In [39, 17] the non-smooth setting is also analyzed. However, only the single direction case has been considered and both algorithms do not achieve the lower bound. More precisely, for convex functions, a complexity of $\mathcal{O}(d^2\varepsilon^{-2})$ is achieved by [39] and $\mathcal{O}(d\log(d)\varepsilon^{-2})$ by [17] while our algorithm gets the optimal dependence. Moreover, note that in [17] the strategy adopted (also called "double smoothing") requires tuning one more sequence of parameters, which is a challenging problem in practice, and only the convex setting is considered. In [39], also the non-convex setting is analyzed by bounding the expected norm of the smoothed gradient. However, they obtain a complexity of $\mathcal{O}(d^3\varepsilon^{-2}h^{-1})$ while our algorithm gets a better dependence on the dimension. In [46], the optimal complexity is obtained for convex functions. However, the author does not analyze the non-convex setting. Moreover, note that, despite the complexity in terms of function evaluations being the same, our algorithm gets a better complexity in terms of the number of iterations since it uses multiple directions (and this is an advantage if we can parallelize the function evaluations). Furthermore, note that the method proposed in [46] can be seen as the special case of Algorithm 1 with $\ell = 1$. In [34] the single direction case is analyzed only in the non-convex setting. The dependence on the dimension of the complexity in the number of function evaluations achieved matches our result in this setting (again, in the number of iterations our method obtains a better dependence).

**Structured Finite-difference.** In [32, 42], authors analyze structured finite differences in both deterministic and stochastic settings. However, only the smooth convex setting is considered. In [12] orthogonal matrices are used to build directions but no analysis is provided. In [25], finite-difference with coordinate directions is analyzed. At every iteration, $d + 1$ function evaluations are performed to compute the estimator and only the smooth setting is considered.

## 3 Main Results

In this section, we analyze Algorithm 1 considering both non-smooth and smooth problems. We present the rates obtained by our method for convex and non-convex settings and compare them with those obtained by state-of-the-art methods. Proofs are provided in Appendix B. In the following, we call *complexity in the number of iterations / function evaluations*, respectively, the number of iterations / function evaluations required to achieve an accuracy $\varepsilon \in (0, 1)$.

### 3.1 Non-smooth Convex Setting

In this section, we provide the main results for non-smooth convex functions. In particular, we will assume that the target function is convex and satisfy the following hypothesis.

**Assumption 1** ($L_0$-Lipschitz continuous). *The function $f$ is $L_0$-Lipschitz continuous; i.e., for some $L_0 > 0$,*

$$(\forall x, y \in \mathbb{R}^d) \qquad |f(x) - f(y)| \leq L_0 \|x - y\|.$$

Note that this assumption implies that also $f_h$ is $L_0$-Lipschitz continuous - see Proposition 1. Moreover, to analyze the algorithm, we will consider the following parameter setting.

**Assumption 2** (Zeroth-order non-smooth convergence conditions). *The step-size sequence $\alpha_k$ and the sequence of smoothing parameters $h_k$ satisfy the following conditions:*

$$\alpha_k \notin \ell^1, \qquad \alpha_k^2 \in \ell^1 \qquad and \qquad \alpha_k h_k \in \ell^1.$$

The assumption above is required to guarantee convergence to a solution. In particular, the first two conditions are common for subgradient method and stochastic gradient descent, while the third condition was already used in structured zeroth-order methods [32, 42] and links the decay of the smoothing parameter with the stepsize's one. An example of $\alpha_k, h_k$ that satisfy Assumption 2 is $\alpha_k = k^{-\theta}$ and $h_k = k^{-\rho}$ with $\theta \in (1/2, 1)$ and $\rho$ s.t. $\theta + \rho > 1$.

We state now the main theorem for non-smooth convex functions.

**Theorem 1** (Non-smooth convex). *Under Assumption 1, assume that $f$ is convex and let $(x_k)_{k \in \mathbb{N}}$ be a sequence generated by Algorithm 1. For every $k \in \mathbb{N}$, denote $A_k = \sum_{i=0}^k \alpha_i$ and set $\bar{x}_k := \sum_{i=0}^k \alpha_i x_i / A_k$. Then*

$$\mathbb{E}[f(\bar{x}_k) - \min f] \leq S_k / A_k \qquad with \qquad S_k := \frac{\|x_0 - x^*\|^2}{2} + c\frac{dL_0^2}{\ell} \sum_{i=0}^k \alpha_i^2 + L_0 \sum_{i=0}^k \alpha_i h_i,$$

*where $c > 0$ is a constant independent of the dimension and $x^*$ is any solution in $\arg\min f$. Moreover, under Assumption 2, we have*

$$\lim_{k \to +\infty} f(x_k) = \min f \quad a.s,$$

*and that there exists a random variable $x^*$ taking values in $\arg\min f$ s.t. $x_k \to x^*$ a.s.*

In the next corollary, we derive explicit rates for specific choices of the parameters.

**Corollary 1.** *Under the assumptions of Theorem 1, let $x^* \in \arg\min f$. Then, the following hold:*

(i) *Let $\theta \in (1/2, 1)$ and $\rho \in \mathbb{R}$ such that $\theta + \rho > 1$. For every $k \in \mathbb{N}$, let $\alpha_k = \alpha(k+1)^{-\theta}$ and $h_k = h(k+1)^{-\rho}$ with $\alpha > 0$ and $h > 0$. Then*

$$\mathbb{E}[f(\bar{x}_k) - \min f] \leq \frac{C}{\alpha k^{1-\theta}} + o\left(\frac{1}{k^{1-\theta}}\right),$$

*for some constant $C$ provided in the proof.*

(ii) *For every $k \in \mathbb{N}$, let $\alpha_k = \alpha$ and $h_k = h$ with $\alpha, h > 0$. Then*

$$\mathbb{E}[f(\bar{x}_k) - \min f] \leq \frac{\|x_0 - x^*\|^2}{2\alpha k} + \frac{cdL_0^2}{\ell}\alpha + L_0 h,$$

*where $c$ is a constant independent of the dimension.*

(iii) *Fix an accuracy $\varepsilon \in (0, 1)$ and let $K \geq 8(cL_0^2\|x_0 - x^*\|^2)(d/\ell)\varepsilon^{-2}$. Set $\alpha_k = \sqrt{\frac{\ell}{d}}\frac{\|x_0 - x^*\|}{\sqrt{2cKL_0}}$, and $h_k = h \leq \varepsilon/(2L_0)$ for every $k \leq K$. Then*

$$\mathbb{E}[f(\bar{x}_K) - \min f] \leq \varepsilon$$

*and the complexity in terms of number of iterations is $\mathcal{O}((d/\ell)\varepsilon^{-2})$.*

**Discussion.** The bound in Theorem 1 depends on the initialization and on an additional quantity that can be interpreted as an approximation error. The latter is composed of two parts. The first one is generated by the approximation of the gradient of the smoothed function; while the second, involving $h_k$, is generated by the smoothing. Since the rate depends on $1/\sum_{i=0}^k \alpha_i$, we would like to choose the stepsize as large as possible. However, to get convergence, we need to make the approximation errors

vanish sufficiently fast. To guarantee this, as we can observe from Theorem 1, we need to impose some conditions on the stepsize $\alpha_k$ and on the smoothing parameter $h_k$ (i.e. Assumption 2), that will slow down the decay of the first term. In Corollary 1, we provide two choices of parameters: the first one satisfies Assumption 2 and ensures convergence; the second one corresponds to constant stepsize and smoothing parameter. For the first choice, we recover the rate of the subgradient method in terms of $k$. In particular, for $\theta$ approaching $1/2$, the convergence rate is arbitrarily close to $O(k^{-1/2})$ and is similar to the one derived in [17, Theorem 2]. The dependence on the dimension depends on the choice of the constant $\alpha$. The optimal dependence is obtained with the choice $\alpha = \sqrt{\ell/d}$. Indeed, in that case, the complexity in the number of iterations is of the order $\mathcal{O}((d/\ell)\varepsilon^{-2})$, which is better than the one achieved by [17, Theorem 2] and [39]. Note that also [46, Corollary 1] and [22, Theorem 2.4] obtain a worse complexity in terms of the number of iterations (since they use a single direction), but the same complexity in the number of function evaluations. Clearly, since multiple directions are used, a single iteration of O-ZD will be more expensive than one iteration of [46, 22]. However, our algorithm is more efficient if the $\ell$ function evaluations required at each iteration can be parallelized. On the more theoretical side, we observe that the advantage of multiple orthogonal directions is in the tighter bounds for the variance of the estimator, namely for $\mathbb{E}[\|g_{(G_k,h_k)}(x_k)\|^2]$ - see [46, Lemma 5] and Lemma 4.

## 3.2 Non-smooth Non-convex Setting

To analyze the non-convex setting, following [39], we provide a bound on the averaged expected square norm of the gradient of the smoothed target. In particular, we use the following notation:

$$\eta_k^{(h)} := \left( \sum_{i=0}^{k} \alpha_i \, \mathbb{E}[\|\nabla f_h(x_i)\|^2] \right) / A_k, \qquad \text{where} \ \ A_k := \sum_{i=0}^{k} \alpha_i.$$

Next, we state the main theorem for the non-convex non-smooth setting.

**Theorem 2** (Non-smooth non-convex). *Under Assumption 1, let $(x_k)_{k\in\mathbb{N}}$ be a sequence generated by Algorithm 1 with, for every $k \in \mathbb{N}$, $h_k = h$ for some $h > 0$. Then*

$$\eta_k^{(h)} \leq S_k/A_k \quad \text{with} \quad S_k := f_h(x_0) - \min f + c \frac{L_0^3 d\sqrt{d}}{\ell} \sum_{i=0}^{k} \frac{\alpha_i^2}{h}.$$

In the next corollary, we derive the rates for specific choices of the parameters.

**Corollary 2.** *Under the assumptions of Theorem 2, the following statements hold.*

*(i) If $\alpha_k = \alpha(k+1)^{-\theta}$ with $\alpha > 0$ and $\theta \in (1/2, 1)$, then*

$$\eta_k^{(h)} \leq C \frac{f_h(x_0) - \min f}{\alpha k^{1-\theta}} + o\left( \frac{1}{k^{1-\theta}} \right),$$

*where $C$ is a constant independent of the dimension.*

*(ii) If $\alpha_k = \alpha$ with $\alpha > 0$ for every $k \in \mathbb{N}$, then*

$$\eta_k^{(h)} \leq \frac{f_h(x_0) - \min f}{\alpha k} + \frac{cL_0^3 d\sqrt{d}}{\ell h}\alpha,$$

*where $c$ is a constant independent of the dimension.*

*(iii) Let $\varepsilon \in (0,1)$, let $K \geq 4(f_h(x_0) - \min f)cL_0^3 d\sqrt{d}\varepsilon^{-2}/(\ell h)$ and choose $\alpha = \sqrt{\frac{(f_h(x_0) - \min f)\ell h}{KcL_0^3 d\sqrt{d}}}$. Then we have that $\eta_K^{(h)} \leq \varepsilon$. Thus, the number of function evaluations required to get a precision $\eta_k^h \leq \varepsilon$ is of the order $\mathcal{O}(d\sqrt{d}h^{-1}\varepsilon^{-2})$.*

Relying on the results in [34], we show that this is related to a precise notion of approximate stationarity. To do so, we need to introduce a definition of subdifferential which is suitable to this setting. As shown in [34] the Clarke subdifferential is not the right notion, and the approximate Goldstein subdifferential should be used instead.

**Definition 1** (Goldstein subdifferential and stationary point). *Under Assumption 1, let $x \in \mathbb{R}^d$ and $h > 0$. The $h$-Goldstein subdifferential of $f$ at $x$ is $\partial_h f(x) := conv(\cup_{y \in \mathbb{B}_h^d(x)} \partial f(y))$ where $\partial f$ is the Clarke subdifferential [13] and $\mathbb{B}_h^d(x)$ is the ball centered in $x$ with radius $h$. For $\varepsilon \in (0,1)$, a point $x \in \mathbb{R}^d$ is a $(h, \varepsilon)$-Goldstein stationary point for the function $f$ if $\min\{\|g\| \mid g \in \partial_h f(x)\} \leq \varepsilon$.*

**Corollary 3.** *Under the same assumptions of Theorem 2, fix $K \in \mathbb{N}$ and let $I$ be a random variable taking values in $\{0, \dots, K-1\}$ such that, for all $i$, $\mathbb{P}[I = i] = \alpha_i / A_{K-1}$. Let also $S_k$ be defined as in Theorem 2. Then*

$$\mathbb{E}_I\left[\min\{\|\eta\|^2 : \eta \in \partial f_h(x_I)\}\right] \leq S_K/A_K.$$

*In the setting of Corollary 2 (iii), we have also that $\mathbb{E}_I\left[\min\{\|\eta\|^2 : \eta \in \partial f_h(x_I)\}\right] \leq \varepsilon$.*

**Discussion.** In Theorem 2, we fix the smoothing of the target, i.e. we consider $h_k$ constant, and we analyze the non-smooth non-convex setting providing a rate on the expected norm of the smoothed gradient. The resulting bound is composed of two parts. The first part is very natural, and due to the functional value at the initialization. The second part is the approximation error. Recall that Assumption 1 holds, and therefore $f_h \leq f + L_0 h$ due to Proposition 1. This suggests taking $h$ as small as possible in order to reduce the gap between $f_h$ and $f$. However, taking $h$ too small would make the approximation error very big. In our analysis, we consider the case with $h$ constant. Moreover, as for the convex case, the speed of the rate depends on $A_k$ and so we would like to take the stepsize as large as possible. But to control the approximation error, we need to assume $\alpha_k^2 \in \ell^1$. In Corollary 2, we consider two choices of stepsize. The first choice satisfies the property of $\alpha_k^2 \in \ell^1$, while the second one analyzes the case of constant step-size. Comparing our rate to the one in [39] we see that we obtain a better dependence on the dimension in the complexity, both in terms of iterations and function evaluations. Our results match the one of [34, Theorem 3.2] in terms of rate and in terms of function evaluations. We get a better dependence on the dimension in the number of iterations. Note again that, despite the complexity in terms of the number of function evaluations being the same, the possibility of parallelization for the function evaluations yields a better result for our method. As for the convex setting, we have a tighter upper-bound on the variance of the estimator of the smoothed gradient with respect to the dimension - see [34, Lemma D.1]. Goldstein stationarity has been used to assess the approximate stationarity for first-order methods as well, see [15]. The latter work shows that a cutting plane algorithm achieves a rate of $\mathcal{O}(d\varepsilon^{-3})$ for Lipschitz functions.

### 3.3 Smooth Convex setting

We consider now the smooth setting, i.e. we assume that the target function satisfies the following hypothesis.

**Assumption 3** ($L_1$-Smooth). *The function $f$ is $L_1$-smooth; i.e. the function $f$ is differentiable and, for some $L_1 > 0$,*

$$(\forall x, y \in \mathbb{R}^d) \qquad \|\nabla f(x) - \nabla f(y)\| \leq L_1 \|x - y\|.$$

This is the standard assumption for analyzing first-order methods and has been used in many other works in the literature for zeroth-order algorithms - see e.g. [39, 17]. As shown in previous works, if $f$ satisfies Assumption 3 then also $f_h$ satisfies it - see Proposition 1. We will consider also the following assumptions on the stepsize and the smoothing in order to guarantee convergence.

**Assumption 4** (Smooth zeroth-order convergence conditions). *The stepsize sequence $(\alpha_k)_{k \in \mathbb{N}}$ and the smoothing sequence $(h_k)_{k \in \mathbb{N}}$ satisfy the following conditions:*

$$\alpha_k \notin \ell^1 \quad and \quad \alpha_k h_k \in \ell^1.$$

*Moreover, $\alpha_k \leq \bar{\alpha} < \ell/dL_1$ for every $k \in \mathbb{N}$.*

Note that this is a weaker version of Assumption 2. Next, we state the main theorem for convex smooth functions.

**Theorem 3** (Smooth convex). *Under Assumptions 3 and 4, let $(x_k)_{k \in \mathbb{N}}$ be a sequence generated by Algorithm 1 and $x^* \in \arg\min f$. For every $k \in \mathbb{N}$, set $A_k = \sum_{i=0}^k \alpha_i$ and $\bar{x}_k = \sum_{i=0}^k \alpha_i x_i / A_k$. Then, for every $k \in \mathbb{N}$,*

$$\mathbb{E}[f(\bar{x}_k) - \min f] \leq \frac{D_k}{A_k} \quad with \quad D_k := \frac{\ell\Delta + d\bar{\alpha}}{2\ell\Delta}\left(S_k + \sum_{i=0}^k \rho_i\left(\sqrt{S_i} + \sum_{j=0}^i \rho_j\right)\right),$$

where $S_k := \|x_0 - x^*\|^2 + \sum_{i=0}^{k} \frac{L_1^2 d^2}{2\ell} \alpha_i^2 h_i^2$, $\rho_k := L_1 d \alpha_k h_k$, and $\Delta := \left( \frac{1}{L_1} - \frac{d}{\ell} \bar{\alpha} \right)$.

**Corollary 4.** *Under the same Assumptions of Theorem 3, the following hold.*

    *(i) If for every $k \in \mathbb{N}$ we set $\alpha_k = \alpha > 0$ and $h_k = h(k+1)^{-\theta}$ for $h > 0$ and $\theta > 1$, then*

$$\mathbb{E}[f(\bar{x}_k) - \min f] \leq \frac{C}{\alpha k},$$

    *where $C$ is a constant provided in the proof. Moreover, if $\alpha < \ell/(2dL_1)$, $\lim\limits_{k \to \infty} f(x_k) = \min f$ a.s. and there exists a random variable $\hat{x}$ taking values in $\arg\min f$ s.t. $x_k \to \hat{x}$ a.s.*

    *(ii) If for every $k \in \mathbb{N}$ we set $\alpha_k = \alpha > 0$ and $0 < h_k \leq h$, then*

$$\mathbb{E}[f(\bar{x}_k) - \min f] \leq \frac{C_1}{k} + C_2 \alpha h + C_3 \alpha^2 h^2 \sqrt{k} + C_4 \alpha^2 h^2 k,$$

    *where $C_1, C_2, C_3$ and $C_4$ are non-negative constants.*

**Discussion.** As in the previous cases, the bound in Theorem 3 is composed by two terms: the error due to the initialization and the one due to the approximation. An important difference with the results in the non-smooth setting is that every term in the approximation error is decreasing with respect to the smoothing parameter $h_k$. This allows obtaining convergence also with the constant step-size scheme, taking $h_k \in \ell^1$. In Corollary 4 (i), we recover the result of [32, Theorem 5.4] with a specific choice of parameters $\alpha, h$ (up to constants). The complexity depends on the choice of $\alpha$. Note that by Assumption 4, $\alpha < \ell/(L_1 d)$ thus the dependence on the dimension in the rate will be at least $d/\ell$. In particular, taking $\alpha = \ell/(2dL_1)$, we obtain the optimal complexity of $\mathcal{O}(d\varepsilon^{-1})$ in terms of function evaluations. This result has a better dependence on the dimension than [39]. In Corollary 4 (ii), the dependence on the dimension in the complexity depends on the choice of $\alpha$ and $h$. Moreover, the rate obtained is equal (up to constants) to the rate obtained in [32] in the same setting, i.e. $\mathcal{O}(1/k)$ (in which we hide the dependence on $d$ and $\ell$). As for [32], for the first setting we can prove the almost sure convergence of the iterates.

### 3.4 Smooth Non-Convex setting

To analyze the smooth non-convex setting, we introduce the following notation:

$$(\forall k \in \mathbb{R}^d) \qquad A_k := \sum_{i=0}^{k} \alpha_i, \qquad \eta_k := \left( \sum_{i=0}^{k} \alpha_i \, \mathbb{E}[\|\nabla f(x_i)\|^2] \right) / A_k.$$

Note that, in comparison with the quantity defined in Section 3.2, here $\eta_k$ is related to the exact objective function $f$ and not to its smoothed version $f_h$. Next, we state the main result for smooth non-convex functions.

**Theorem 4** (Smooth non-convex). *Suppose that Assumption 3 holds and assume that, for every $k \in \mathbb{N}$, $\alpha_k \leq \bar{\alpha} < \ell/(2dL_1)$. Let $(x_k)_{k \in \mathbb{N}}$ be a sequence generated by Algorithm 1. Then*

$$\eta_k \leq \frac{1}{\Delta A_k} \left( f(x_0) - \min f + \frac{L_1^2 d^2}{8} \sum_{i=0}^{k} \alpha_i h_i^2 + \frac{L_1^3 d^2}{4\ell} \sum_{i=0}^{k} \alpha_i^2 h_i^2 \right), \qquad \Delta := \left( \frac{1}{2} - \frac{L_1 d}{\ell} \bar{\alpha} \right).$$

**Corollary 5.** *Under the assumptions of Theorem 4, the following hold.*

    *(i) If $\alpha_k = \alpha \leq \bar{\alpha}$ and $h_k = hk^{-\theta}$ with $h > 0$ and $\theta > 1$, then*

$$\eta_k \leq \left[ \frac{f(x_0) - \min f}{\Delta \alpha} + \frac{C_1 d^2 h^2}{\Delta} + \frac{C_2 \alpha h^2 d^2}{\Delta \ell} \right] \cdot \frac{1}{k},$$

    *where $C_1$ and $C_2$ are constants provided in the proof.*

    *(ii) If $\alpha_k = \alpha \leq \bar{\alpha}$ and $h_k = h > 0$, then*

$$\eta_k \leq \frac{f(x_0) - \min f}{\Delta \alpha k} + \frac{C_1 d^2 h^2}{\Delta} + \frac{C_2 \alpha h^2 d^2}{\Delta \ell},$$

    *where $C_1$ and $C_2$ are constants provided in the proof.*

**Discussion.** As for the convex case, every term in the approximation error depends on the smoothing parameter $h_k$. In Corollary 5 (i), we take constant step-size and $h_k \in \ell^1$. With this choice of parameters, we get a rate of $\mathcal{O}(1/k)$ which matches with the result obtained by [39]. The dependence on the dimension depends on the choice of $\alpha$ and $h$. Note that $\alpha < \ell/(2dL_1)$, thus taking $h = \mathcal{O}(1/d)$, in the rate we get a dependence on the dimension of $d/\ell$. Taking for instance $\alpha = \ell/(3dL_1)$ and $h = \mathcal{O}(1/d)$, we get a complexity of $\mathcal{O}(d\varepsilon^{-1})$ in terms of function evaluations.

## 4 Numerical Results

In this section, we provide some numerical experiments to assess the performances of our algorithm. We consider two target functions: a convex smooth one and a convex non-smooth one. Details on target functions and parameters of the algorithms are reported in Appendix C. To report our findings, we run the experiments 10 times and provide the mean and standard deviation of the results.

**How to choose the number of directions?** In these experiments, we set a fixed budget of $4000$ function evaluations and we consider $d = 50$. We investigate how the performance of Algorithm 1 changes as the value of $\ell$ increases. In Figure 1, we observe the mean sequence $f(x_k) - f(x^*)$ after each function evaluation. If $\ell > 1$, then the target function values are repeated $2\ell$ times, since we need to perform $2\ell$ function evaluations to do one iteration. For a sufficiently large budget, increasing the number of directions $\ell$ leads to better results compared to using a single direction in both smooth and non-smooth settings.

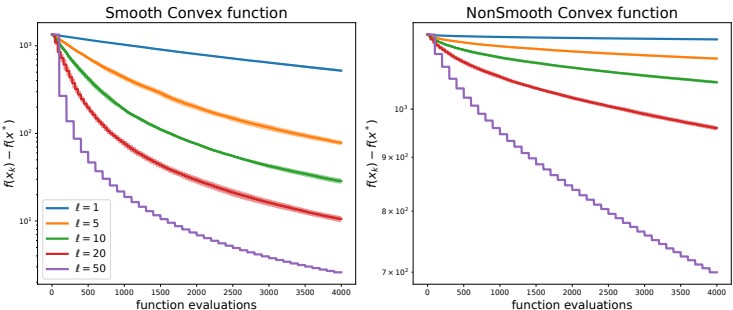

Figure 1: From left to right, function values per function evaluation in optimizing smooth and non-smooth target functions with different numbers of directions.

**Comparison with finite-difference methods.** Now, we compare Algorithm 1 with other finite-difference methods. More precisely, we consider finite differences with single (and multiple) Gaussian (and spherical) directions. The budget of function evaluations is 1000 and the ambient dimension is $d = 10$. For multiple direction methods, we fix the number of directions $\ell = d$. Further experiments are provided in Appendix F.

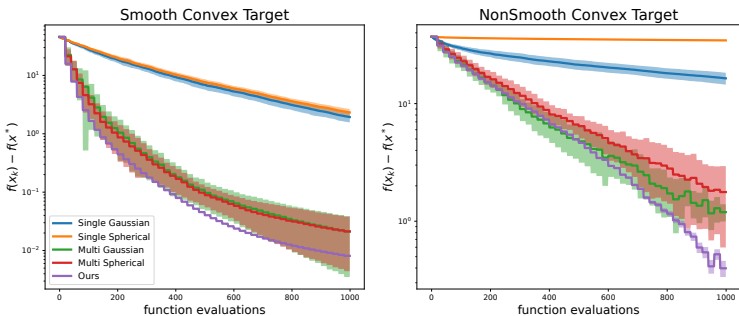

Figure 2: From left to right, function values per function evaluation in optimizing smooth and non-smooth convex functions with different finite-difference algorithms.

In Figure 2, we plot the sequence $f(x_k) - f(x^*)$ with respect to the number of function evaluations. While in terms of rates and complexity the different algorithms are the same, Algorithm 1 shows better performances than random directions approaches, and we believe this is due to the use of structured (i.e. orthogonal) directions. Indeed, orthogonal directions yield a better approximation of first-order information with respect to other methods. The practical advantages of structured directions were already observed in [32, 42, 5, 12] and these experiments confirm that the good practical behavior holds even in the nonsmooth setting.

## 5 Conclusion

We introduced and analyzed O-ZD a zeroth-order algorithm for non-smooth zeroth-order optimization. We analyzed the algorithm and derived rates for non-smooth and smooth functions. This work opens different research directions. An interesting one would be the introduction of a learning procedure for the orthogonal directions. Such an approach could have significant practical applications.

## Acknowledgments and Disclosure of Funding

This project has been supported by the TraDE-OPT project, which received funding from the European Union's Horizon 2020 research and innovation program under the Marie Skłodowska-Curie grant agreement No 861137. L. R. and M. R. acknowledge the financial support of the European Research Council (grant SLING 819789), the AFOSR projects FA9550-18-1-7009 (European Office of Aerospace Research and Development), the EU H2020-MSCA-RISE project NoMADS - DLV-777826, and the Center for Brains, Minds and Machines (CBMM), funded by NSF STC award CCF-1231216. S. V. and L. R. acknowledge the support of the AFOSR project FA8655-22-1-7034. The research by S. V. and C. M. has been supported by the MIUR Excellence Department Project awarded to Dipartimento di Matematica, Università di Genova, CUP D33C23001110001. S. V. and C. M. are members of the Gruppo Nazionale per l'Analisi Matematica, la Probabilità e le loro Applicazioni (GNAMPA) of the Istituto Nazionale di Alta Matematica (INdAM). This work represents only the view of the authors. The European Commission and the other organizations are not responsible for any use that may be made of the information it contains.

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

# A Auxiliary Results

In this appendix, we state and collect lemmas and propositions required to prove the main results.

**Notation.** In the following sections, we denote with $\mathcal{F}_k$ the filtration $\sigma(G_1, \cdots, G_{k-1})$. Moreover, to simplify the notation, we define $g_k$ as the gradient surrogate in eq.(2) at time-step $k$ i.e. $g_k := g_{(G_k, h_k)}(x_k)$ and $g(\cdot) := g_{(G,h)}(\cdot)$ for an arbitrary $G \in O(d)$ and $h > 0$. We denote the normalized Haar measure [37] by $\mu$. We define the unit ball $\mathbb{B}^d$ and the unit sphere $\mathbb{S}^{d-1}$ as follow

$$\mathbb{B}^d := \{v \in \mathbb{R}^d \mid \|v\| \le 1\} \qquad \text{and} \qquad \mathbb{S}^{d-1} := \{v \in \mathbb{R}^d \mid \|v\| = 1\}.$$

We denote by $\sigma$ and $\sigma_N$ the spherical measure and the normalized spherical measure on $\mathbb{S}^{d-1}$, respectively. Moreover, we denote with $I_{d,\ell} \in \mathbb{R}^{d \times \ell}$ the (truncated) identity matrix.

**Lemma 2.** *Let $\beta(\mathbb{S}^{d-1})$ be the surface area of $\mathbb{S}^{d-1}$ and let $I \in \mathbb{R}^{d \times d}$ be the identity matrix. Then,*

$$\int_{\mathbb{S}^{d-1}} vv^\mathsf{T} \, d\sigma(v) = \frac{\beta(\mathbb{S}^{d-1})}{d} I.$$

*Proof.* This result is proved in [21, Lemma 7.3, point (b)]. □

**Lemma 3.** *Let $\phi : \mathbb{R}^d \to \mathbb{R}$ be a L-Lipschitz function. If $u$ is uniformly distributed on $\mathbb{S}^{d-1}$, then*

$$(\mathbb{E}[\phi(u) - \mathbb{E}[\phi(u)]])^2 \le c\frac{L^2}{d},$$

*for some numerical constant $c > 0$.*

*Proof.* The proof follows the same line as [46, Lemma 9]. □

## A.1 Smoothing Lemma & Properties

In this appendix, we provide the proof of the Smoothing Lemma (i.e. Lemma 1).

**Proof of Smoothing Lemma.** By eq. (2),

$$\mathbb{E}_G[g_{(G,h)}(x)] = \frac{d}{\ell} \sum_{i=1}^{\ell} \int_{O(d)} \frac{f(x + hGe_i) - f(x - hGe_i)}{2h} Ge_i \, d\mu(G).$$

By [37, Theorem 3.7],

$$\mathbb{E}_G[g_{(G,h)}(x)] = \frac{d}{2\ell h} \sum_{i=1}^{\ell} \int_{\mathbb{S}^{d-1}} (f(x + hv^{(i)}) - f(x - hv^{(i)}))v^{(i)} \, d\sigma_N(v^{(i)}).$$

Since $v^{(i)}$ is uniformly distributed on the sphere, which is symmetric with respect to the origin, we have

$$\mathbb{E}_G[g_{(G,h)}(x)] = \frac{d}{\ell h} \sum_{i=1}^{\ell} \int_{\mathbb{S}^{d-1}} f(x + hv^{(i)})v^{(i)} \, d\sigma_N(v^{(i)}).$$

As a consequence of Stokes' Theorem (details in [18, Lemma 1] and [1, Theorem A8.8]), we get

$$\mathbb{E}[g_{(G,h)}(x)] = \frac{1}{\ell} \sum_{i=1}^{\ell} \nabla f_h(x) \qquad \text{with} \qquad f_h(x) := \frac{1}{\text{vol}(\mathbb{B}^d)} \int_{\mathbb{B}^d} f(x + hu) \, du.$$

Rearranging terms, we get the claim. □

**Proposition 1** (Smoothing properties)**.** *Let $f_h$ be the smooth approximation of $f$ defined in eq. (4). Then the following hold:*
*If $f$ is convex then $f_h$ is convex and, for every $x \in \mathbb{R}^d$,*

$$f(x) \le f_h(x).$$

*If $f$ is $L_0$-Lipschitz continuous - i.e. $\forall x, y \in \mathbb{R}^d$, $|f(x) - f(y)| \leq L_0 \|x - y\|$, then $f_h$ is $L_0$-Lipschitz continuous, differentiable and for every $x, y \in \mathbb{R}^d$*

$$\|\nabla f_h(x) - \nabla f_h(y)\| \leq \frac{L_0 \sqrt{d}}{h} \|x - y\| \quad and \quad f_h(x) \leq f(x) + L_0 h.$$

*If $f$ is $L_1$-smooth - i.e. $f$ is differentiable and $\forall x, y \in \mathbb{R}^d$, $\|\nabla f(x) - \nabla f(y)\| \leq L_1 \|x - y\|$ then $f_h$ is $L_1$-smooth and for every $x \in \mathbb{R}^d$,*

$$\|\nabla f_h(x) - \nabla f(x)\| \leq \frac{hdL_1}{2} \quad and \quad f_h(x) \leq f(x) + \frac{L_1}{2} h^2.$$

*Proof.* These are standard results proposed and proved in different works - see for example [16, Lemma 8],[21, Proposition 7.5],[34, Proposition 2.2],[49]. $\qquad\square$

**Lemma 4** (Approximation Error). *Let $g(\cdot)$ be the surrogate defined in eq. (2) for arbitrary $h > 0$ and $G \in O(d)$. Then the following hold:*

(i) *If $f$ is $L_0$-Lipschitz (see Assumption 1), then, for every $x \in \mathbb{R}^d$,*

$$\mathbb{E}_G[\|g(x)\|^2] \leq 2c \frac{dL_0^2}{\ell},$$

*where $c$ is a numerical constant.*

(ii) *If $f$ is $L_1$-smooth (see Assumption 3), then, for every $x \in \mathbb{R}^d$,*

$$E_G[\|g(x)\|^2] \leq \frac{2d}{\ell} \|\nabla f(x)\|^2 + \frac{L_1^2 d^2}{2\ell} h^2.$$

*Proof.* Note that, since directions are orthogonal, we have

$$\mathbb{E}_G[\|g(x)\|^2] = \frac{d^2}{4\ell^2 h^2} \sum_{i=1}^{\ell} \mathbb{E}_G[(f(x + hGe_i) - f(x - hGe_i))^2 \|Ge_i\|^2].$$

By [37, Theorem 3.7],

$$\mathbb{E}_G[\|g(x)\|^2] = \frac{d^2}{4\ell^2 h^2} \sum_{i=1}^{\ell} \mathbb{E}_{v_i}[(f(x + hv^{(i)}) - f(x - hv^{(i)}))^2 \|v^{(i)}\|^2], \qquad (5)$$

where each $v^{(i)}$ is uniformly distributed on $\mathbb{S}^{d-1}$.

(i): Set $\gamma = \mathbb{E}_{v^{(i)}}[f(x + hv^{(i)})]$ for every $i$ (this expectation does not depend on $i$). Then

$$\mathbb{E}_G[\|g(x)\|^2] = \frac{d^2}{4\ell^2 h^2} \sum_{i=1}^{\ell} \mathbb{E}_{v^{(i)}}[(f(x + hv^{(i)}) - f(x - hv^{(i)}) + \gamma - \gamma)^2 \|v^{(i)}\|^2]$$

$$= \frac{d^2}{4\ell^2 h^2} \sum_{i=1}^{\ell} \mathbb{E}_{v^{(i)}}[((f(x + hv^{(i)}) - \gamma) - (f(x - hv^{(i)}) - \gamma))^2 \|v^{(i)}\|^2]$$

$$\leq \frac{d^2}{2\ell^2 h^2} \sum_{i=1}^{\ell} \mathbb{E}_{v^{(i)}}[((f(x + hv^{(i)}) - \gamma)^2 + (f(x - hv^{(i)}) - \gamma)^2) \|v^{(i)}\|^2]$$

$$= \frac{d^2}{2\ell^2 h^2} \sum_{i=1}^{\ell} \Big[ \mathbb{E}_{v^{(i)}}[(f(x + hv^{(i)}) - \gamma)^2 \|v^{(i)}\|^2]$$

$$+ \mathbb{E}_{v^{(i)}}[(f(x - hv^{(i)}) - \gamma)^2 \|v^{(i)}\|^2] \Big].$$

Since $v^{(i)}$ is uniformly distributed on $\mathbb{S}^{d-1}$, it satisfies $\|v^{(i)}\|^2 = 1$ and by symmetry we have

$$\mathbb{E}_G[\|g(x)\|^2] \leq \frac{d^2}{\ell^2 h^2} \sum_{i=1}^{\ell} \mathbb{E}_{v^{(i)}}[(f(x + hv^{(i)}) - \gamma)^2].$$

The definition of $\gamma$ yields

$$\mathbb{E}_G[\|g(x)\|^2] \le \frac{d^2}{\ell^2 h^2} \sum_{i=1}^{\ell} \mathbb{E}_{v^{(i)}}[((f(x+hv^{(i)}) - \gamma)^2]$$

$$= \frac{d^2}{\ell^2 h^2} \sum_{i=1}^{\ell} \mathbb{E}_{v^{(i)}}[(f(x+hv^{(i)}) - \mathbb{E}_{v^{(i)}}[f(x+hv^{(i)})])^2].$$

The claim follows by Lemma 3 and the fact that $f(x + hv^{(i)})$ is $hL_0$-Lipschitz continuous w.r.t to $v^{(i)}$.

$(ii)$: Equation (5) yields

$$\mathbb{E}_G[\|g(x)\|^2] = \frac{d^2}{4\ell^2 h^2} \sum_{i=1}^{\ell} \mathbb{E}_{v^{(i)}}[(f(x+hv^{(i)}) - f(x-hv^{(i)}) - f(x) + f(x))^2 \|v^{(i)}\|^2]$$

$$\le \frac{d^2}{2\ell^2 h^2} \sum_{i=1}^{\ell} \Big[ \mathbb{E}_{v^{(i)}}[(f(x+hv^{(i)}) - f(x))^2 \|v^{(i)}\|^2]$$

$$+ \mathbb{E}_{v^{(i)}}[(f(x-hv^{(i)}) - f(x))^2 \|v^{(i)}\|^2] \Big]$$

$$= \frac{d^2}{\ell^2 h^2} \sum_{i=1}^{\ell} \mathbb{E}_{v^{(i)}}[(f(x+hv^{(i)}) - f(x))^2],$$

where the last equation follows by symmetry. Adding and subtracting $\langle \nabla f(x), hv^{(i)} \rangle$ we derive

$$\mathbb{E}_G[\|g(x)\|^2] \le \frac{d^2}{\ell^2 h^2} \sum_{i=1}^{\ell} \mathbb{E}_{v^{(i)}} \left[ \left( f(x+hv^{(i)}) - f(x) - \langle \nabla f(x), hv^{(i)} \rangle + \langle \nabla f(x), hv^{(i)} \rangle \right)^2 \right]$$

$$\le \frac{2d^2}{\ell^2 h^2} \sum_{i=1}^{\ell} \left( \mathbb{E}_{v^{(i)}} \left[ \left( f(x+hv^{(i)}) - f(x) - \langle \nabla f(x), hv^{(i)} \rangle \right)^2 \right] \right.$$

$$+ \mathbb{E}_{v^{(i)}} \left[ \left( \langle \nabla f(x), hv^{(i)} \rangle \right)^2 \right] \bigg).$$

Denote by $\beta(\mathbb{S}^{d-1})$ the surface area of $\mathbb{S}^{d-1}$. The Descent Lemma [41] implies

$$\mathbb{E}_G[\|g(x)\|^2] \le \frac{2d^2}{\ell^2 h^2} \sum_{i=1}^{\ell} \left[ \left( \frac{L_1^2}{4} h^4 \right) + \mathbb{E} \left[ \left( \langle \nabla f(x), hv^{(i)} \rangle \right)^2 \right] \right]$$

$$= \frac{L_1^2 d^2}{2\ell} h^2 + \frac{2d^2}{\ell^2 h^2} \sum_{i=1}^{\ell} \mathbb{E} \left[ \left( \langle \nabla f(x), hv^{(i)} \rangle \right)^2 \right]$$

$$= \frac{L_1^2 d^2}{2\ell} h^2 + \frac{2d^2}{\ell^2 \beta(\mathbb{S}^{d-1})} \sum_{i=1}^{\ell} \int_{\mathbb{S}^{d-1}} \nabla f(x)^\intercal v^{(i)} v^{(i)\intercal} \nabla f(x) \, d\sigma(v).$$

By Lemma 2, we get the claim. Indeed,

$$\mathbb{E}_G[\|g(x)\|^2] \le \frac{L_1^2 d^2}{2\ell} h^2 + \frac{2d^2}{\ell^2 \beta(\mathbb{S}^{d-1})} \sum_{i=1}^{\ell} \left( \frac{\beta(\mathbb{S}^{d-1})}{d} \|\nabla f(x)\|^2 \right)$$

$$= \frac{2d}{\ell} \|\nabla f(x)\|^2 + \frac{L_1^2 d^2}{2\ell} h^2.$$

$\square$

## A.2 Auxiliary results and proofs for the nonsmooth setting, convex, and nonconvex.

In this subsection, for every $k$, we will denote by $\mathcal{F}_k$ the $\sigma$-algebra $\sigma(G_0, \ldots, G_{k-1})$.

**Lemma 5.** *Let $f : \mathbb{R}^d \to \mathbb{R}$ be a lower semi-continuous function and denote with $S = \arg\min f$ and $f^* = \min f$. Then,*

$$\begin{cases} \text{(A)} & \forall x^* \in S, \ \exists \lim_{k} \|x_k - x^*\| \\ \text{(B)} & \liminf_{k} f(x_k) = f^* \end{cases} \implies \exists x_\infty \in S \quad s.t. \quad x_k \to x_\infty.$$

*Proof.* Since (B) holds, we have that exists $(x_{k_j})_{j \in \mathbb{N}}$ subsequence of $(x_k)_{k \in \mathbb{N}}$ such that $f(x_{k_j}) \to f^*$. Since $S \neq \emptyset$ and (A) we have that

$$\exists x^* \in S \quad \text{and} \quad \exists \lim_{k} \|x_k - x^*\|.$$

Thus, the sequence $(x_k)_{k \in \mathbb{N}}$ is bounded and, therefore, also $(x_{k_j})_{j \in \mathbb{N}}$ is bounded. Taking a convergent subsequence $(x_{k_{j_n}})_{n \in \mathbb{N}}$ of $(x_{k_j})_{j \in \mathbb{N}}$, we have that exists $x_\infty$ s.t.

$$x_{k_{j_n}} \to x_\infty.$$

Since $f$ is assumed to be lower semi-continuous, we have that

$$f(x_\infty) \leq \liminf_{n} f(x_{k_{j_n}}) = f^* = \lim_{j} f(x_{k_j}).$$

Thus, we have that $x_\infty \in S$ which implies, by (A), that

$$\exists \lim_{k} \|x_k - x_\infty\| \quad \text{and} \quad \lim_{n} \|x_{k_{j_n}} - x_\infty\| = 0.$$

Hence, since $x_{k_{j_n}}$ is a subsequence of $x_k$,

$$\lim_{k} \|x_k - x_\infty\| = 0,$$

and, therefore, $x_k \to x_\infty \in S$. $\qquad\square$

**Lemma 6** (Convergence: convex non-smooth). *Assume that $f$ is convex and $L_0$ Lipschitz continuous. Let $(x_k)_{k \in \mathbb{N}}$ be the sequence generated by Algorithm 1 and let $x^* \in \arg\min f$. Then, for every $k \in \mathbb{N}$, the following inequality holds:*

$$\mathbb{E}[\|x_{k+1} - x^*\|^2 | \mathcal{F}_k] - \|x_k - x^*\|^2 + 2\alpha_k(f(x_k) - f(x^*)) \leq 2c\frac{L_0^2 d}{\ell}\alpha_k^2 + 2L_0\alpha_k h_k,$$

*where $c$ is some non-negative constant independent from the dimension. Moreover, if the stepsizes satisfy Assumption 2, we have*

$$\lim_{k \to +\infty} f(x_k) = f(x^*) \quad a.s,$$

*and there exists a random variable $\hat{x}$ taking values in in $\arg\min f$ such that $x_k \to \hat{x}$ a.s.*

*Proof.* Let $k \in \mathbb{N}$. By Algorithm 1,

$$\|x_{k+1} - x^*\|^2 - \|x_k - x^*\|^2 = \alpha_k^2 \|g_k\|^2 - 2\alpha_k \langle g_k, x_k - x^* \rangle. \tag{6}$$

Since $f_{h_k}$ is convex by Proposition 1 and $\mathbb{E}[g_k | \mathcal{F}_k] = \nabla f_{h_k}(x_k)$ (see Lemma 1), we have

$$-\langle \nabla f_{h_k}(x_k), x_k - x^* \rangle \leq f_{h_k}(x^*) - f_{h_k}(x_k).$$

Thus, taking the conditional expectation with respect to $\mathcal{F}_k$, by Lemma 4, we get,

$$\mathbb{E}[\|x_{k+1} - x^*\|^2 | \mathcal{F}_k] - \|x_k - x^*\|^2 \leq \underbrace{2c\frac{L_0^2 d}{\ell}\alpha_k^2}_{=:C_k} - 2\alpha_k(f_{h_k}(x_k) - f_{h_k}(x^*)).$$

Then, by Proposition 1,

$$\mathbb{E}[\|x_{k+1} - x^*\|^2 | \mathcal{F}_k] - \|x_k - x^*\|^2 \leq C_k - 2\alpha_k(f(x_k) - f(x^*)) + 2L_0\alpha_k h_k.$$

Next suppose that Assumption 2 holds. Rearranging the terms,

$$\mathbb{E}[\|x_{k+1} - x^*\|^2 | \mathcal{F}_k] - \|x_k - x^*\|^2 + 2\alpha_k(f(x_k) - f(x^*)) \leq C_k + 2L_0\alpha_k h_k,$$

with $C_k \in \ell^1$ and $\alpha_k h_k \in \ell^1$. Therefore, Robbins-Siegmund Theorem [43] implies that $(\|x_k - x^*\|)_{k \in \mathbb{N}}$ is a.s. convergent, $\alpha_k(f(x_k) - f(x^*)) \in \ell^1$ a.s. and thus, since $\alpha_k \notin \ell^1$,

$$\liminf_{k \to \infty} f(x_k) = f(x^*) \quad \text{a.s.} \tag{7}$$

We derive from [32, Lemma 9.9] and Lemma 5 that there exists a random variable $\hat{x}$ taking values in $\arg\min f$ such that $x_k \to \hat{x}$ a.s. Finally, continuity of $f$ yields that $\lim_{k} f(x_k) = f(x_*)$ a.s. $\qquad\square$

In the next Lemma, to derive bounds on function values, we study the sequence $(f_{h_k}(x_{k+1}) - f_{h_k}(x_k))_{k\in\mathbb{N}}$. It is the difference between the smoothed function at iteration $k$ evaluated at $x_k$ and at $x_{k+1}$. It corresponds to the function value decrease between the iterations $k + 1$ and $k$ if $h_k$ is constant.

**Lemma 7** (Function Value decrease: nonconvex non-smooth setting)**.** *Under Assumption 1, let* $(x_k)_{k\in\mathbb{N}}$ *be the sequence generated by Algorithm 1. Then,*

$$\mathbb{E}[f_{h_k}(x_{k+1})|\mathcal{F}_k] - f_{h_k}(x_k) \leq -\alpha_k\|\nabla f_{h_k}(x_k)\|^2 + c\frac{L_0^3 d\sqrt{d}}{\ell}\frac{\alpha_k^2}{h_k},$$

*where $c$ is a numerical constant.*

*Proof.* By Lemma 1, we have that $f_{h_k}$ is $L_0\sqrt{d}/h_k$-smooth. Thus, by the Descent Lemma [41],

$$f_{h_k}(x_{k+1}) - f_{h_k}(x_k) \leq -\alpha_k \langle\nabla f_{h_k}(x_k), g_k\rangle + \frac{L_0\sqrt{d}}{2h_k}\alpha_k^2\|g_k\|^2.$$

Taking the conditional expectation with respect to $\mathcal{F}_k$,

$$\mathbb{E}[f_{h_k}(x_{k+1})|\mathcal{F}_k] - f_{h_k}(x_k) \leq -\alpha_k\|\nabla f_{h_k}(x_k)\|^2 + \frac{L_0\sqrt{d}}{2h_k}\alpha_k^2\,\mathbb{E}[\|g_k\|^2|\mathcal{F}_k]. \tag{8}$$

The claim follows from Lemma 4. $\qquad\square$

### A.3 Auxiliary results for smooth setting.

**Lemma 8** (Function value decrease: convex smooth setting)**.** *Under Assumption 3 , let $(x_k)_{k\in\mathbb{N}}$ be the sequence generated by Algorithm 1. Then the following holds:*

$$\mathbb{E}[f(x_{k+1})|\mathcal{F}_k] - f(x_k) \leq -\alpha_k\Big(\frac{1}{2} - \frac{L_1 d}{\ell}\alpha_k\Big)\|\nabla f(x_k)\|^2 + \frac{L_1^2 d^2\alpha_k h_k^2}{8} + \frac{L_1^3 d^2}{4\ell}\alpha_k^2 h_k^2.$$

*Proof.* By the Descent Lemma [41] and Algorithm 1,

$$f(x_{k+1}) - f(x_k) \leq -\alpha_k \langle\nabla f(x_k), g_k\rangle + \frac{L_1}{2}\alpha_k^2\|g_k\|^2.$$

Taking the conditional expectation and by Lemma 4,

$$\mathbb{E}[f(x_{k+1})|\mathcal{F}_k] - f(x_k) \leq -\alpha_k \langle\nabla f(x_k), \nabla f_{h_k}(x_k)\rangle + \frac{L_1}{2}\alpha_k^2\Big[\frac{2d}{\ell}\|\nabla f(x_k)\|^2 + \frac{L_1^2 d^2}{2\ell}h_k^2\Big].$$

Adding and subtracting $\nabla f(x_k)$,

$$\begin{aligned}\mathbb{E}[f(x_{k+1})|\mathcal{F}_k] - f(x_k) &\leq -\alpha_k \langle\nabla f(x_k), \nabla f_{h_k}(x_k) - \nabla f(x_k)\rangle - \alpha_k\|\nabla f(x_k)\|^2 \\ &\quad + \frac{L_1}{2}\alpha_k^2\Big[\frac{2d}{\ell}\|\nabla f(x_k)\|^2 + \frac{L_1^2 d^2}{2\ell}h_k^2\Big].\end{aligned}$$

By Cauchy-Schwarz inequality and Proposition 1,

$$\begin{aligned}\mathbb{E}[f(x_{k+1})|\mathcal{F}_k] - f(x_k) &\leq \alpha_k\Big(\frac{L_1 d}{2}h_k\Big)\|\nabla f(x_k)\| - \alpha_k\|\nabla f(x_k)\|^2 \\ &\quad + \frac{L_1}{2}\alpha_k^2\Big[\frac{2d}{\ell}\|\nabla f(x_k)\|^2 + \frac{L_1^2 d^2}{2\ell}h_k^2\Big].\end{aligned}$$

By Young's inequality,

$$\begin{aligned}\mathbb{E}[f(x_{k+1})|\mathcal{F}_k] - f(x_k) &\leq \frac{L_1^2 d^2\alpha_k h_k^2}{8} + \frac{\alpha_k}{2}\|\nabla f(x_k)\|^2 - \alpha_k\|\nabla f(x_k)\|^2 \\ &\quad + \frac{L_1}{2}\alpha_k^2\Big[\frac{2d}{\ell}\|\nabla f(x_k)\|^2 + \frac{L_1^2 d^2}{2\ell}h_k^2\Big]. \\ &= -\alpha_k\Big(\frac{1}{2} - \frac{L_1 d}{\ell}\alpha_k\Big)\|\nabla f(x_k)\|^2 + \frac{L_1^2 d^2\alpha_k h_k^2}{8} + \frac{L_1^3 d^2}{4\ell}\alpha_k^2 h_k^2.\end{aligned}$$

This concludes the proof. $\qquad\square$

**Lemma 9** (Convergence in smooth setting). *Let $(x_k)_{k \in \mathbb{N}}$ be the sequence generated by Algorithm 1 and let $x^* \in \arg\min_{x \in \mathbb{R}^d} f(x)$. Then, under Assumption 3, the following inequality holds*

$$\mathbb{E}[\|x_{k+1} - x^*\|^2 | \mathcal{F}_k] - \|x_k - x^*\|^2 \leq \frac{2d}{\ell}\alpha_k^2\|\nabla f(x_k)\|^2 + \frac{L_1^2 d^2}{2\ell}\alpha_k^2 h_k^2$$
$$+ L_1 d\alpha_k h_k \|x_k - x^*\| - 2\alpha_k \langle \nabla f(x_k), x_k - x^* \rangle.$$

*Moreover, if $f$ is convex, Assumption 4 holds and $\alpha_k \leq \bar{\alpha} < \ell/(2dL_1)$. Then*

- *$(\alpha_k\|\nabla f(x_k)\|^2)_{k \in \mathbb{N}} \in \ell^1$ a.s.*

- *$(\|x_k - x^*\|)_{k \in \mathbb{N}}$ is a.s. convergent.*

- *$(\alpha_k(f(x_k) - f(x^*)))_{k \in \mathbb{N}} \in \ell^1$ a.s.*

- *there exists a random variable $\hat{x}$ taking values in $\arg\min f$ such that $x_k \to \hat{x}$ a.s. and $\lim_{k \to \infty} f(x_k) = \min f$.*

*Proof.* We have

$$\|x_{k+1} - x^*\|^2 - \|x_k - x^*\|^2 = \alpha_k^2\|g_k\|^2 - 2\alpha_k \langle g_k, x_k - x^* \rangle.$$

Taking the conditional expectation,

$$\mathbb{E}[\|x_{k+1} - x^*\|^2 | \mathcal{F}_k] - \|x_k - x^*\|^2 = \alpha_k^2 \mathbb{E}[\|g_k\|^2 | \mathcal{F}_k] - 2\alpha_k \langle \nabla f_{h_k}(x_k), x_k - x^* \rangle.$$

For every $k$, set $u_k = \|x_k - x^*\|$. By Lemma 4,

$$\mathbb{E}[u_{k+1}^2 | \mathcal{F}_k] - u_k^2 \leq \frac{2d}{\ell}\alpha_k^2\|\nabla f(x_k)\|^2 + \frac{L_1^2 d^2}{2\ell}\alpha_k^2 h_k^2 - 2\alpha_k \langle \nabla f_{h_k}(x_k), x_k - x^* \rangle.$$

Note that

$$-2\alpha_k \langle \nabla f_{h_k}(x_k), x_k - x^* \rangle = 2\alpha_k \langle \nabla f_{h_k}(x_k) - \nabla f(x_k), x^* - x_k \rangle - 2\alpha_k \langle \nabla f(x_k), x_k - x^* \rangle.$$

Thus,

$$\mathbb{E}[u_{k+1}^2 | \mathcal{F}_k] - u_k^2 \leq \frac{2d}{\ell}\alpha_k^2\|\nabla f(x_k)\|^2 + \frac{L_1^2 d^2}{2\ell}\alpha_k^2 h_k^2$$
$$+ 2\alpha_k \langle \nabla f_{h_k}(x_k) - \nabla f(x_k), x^* - x_k \rangle - 2\alpha_k \langle \nabla f(x_k), x_k - x^* \rangle.$$

By the Cauchy-Schwarz inequality,

$$\mathbb{E}[u_{k+1}^2 | \mathcal{F}_k] - u_k^2 \leq \frac{2d}{\ell}\alpha_k^2\|\nabla f(x_k)\|^2 + \frac{L_1^2 d^2}{2\ell}\alpha_k^2 h_k^2$$
$$+ 2\alpha_k\|\nabla f_{h_k}(x_k) - \nabla f(x_k)\|u_k - 2\alpha_k \langle \nabla f(x_k), x_k - x^* \rangle.$$

The first claim follows from Proposition 1. By Proposition 1 and Young's inequality with parameter $\tau_k = \alpha_k h_k$, we get

$$\mathbb{E}[u_{k+1}^2 | \mathcal{F}_k] - u_k^2 \leq \frac{2d}{\ell}\alpha_k^2\|\nabla f(x_k)\|^2 + \frac{L_1^2 d^2}{2\ell}\alpha_k^2 h_k^2$$
$$+ \frac{L_1 d}{2\tau_k}\alpha_k^2 h_k^2 + \frac{L_1 d\tau_k}{2}u_k^2 - 2\alpha_k \langle \nabla f(x_k), x_k - x^* \rangle.$$
$$= \frac{2d}{\ell}\alpha_k^2\|\nabla f(x_k)\|^2 + \frac{L_1^2 d^2}{2\ell}\alpha_k^2 h_k^2 \qquad (9)$$
$$+ \frac{L_1 d}{2}\alpha_k h_k + \frac{L_1 d}{2}\alpha_k h_k u_k^2$$
$$- 2\alpha_k \langle \nabla f(x_k), x_k - x^* \rangle.$$

Since $f$ is convex, by Baillon-Haddad Theorem [3], we derive that

$$\mathbb{E}[u_{k+1}^2 | \mathcal{F}_k] - u_k^2 \leq -2\Big(\frac{1}{L_1} - \frac{d}{\ell}\alpha_k\Big)\alpha_k\|\nabla f(x_k)\|^2 + \frac{L_1^2 d^2}{2\ell}\alpha_k^2 h_k^2$$
$$+ \frac{L_1 d}{2}\alpha_k h_k + \frac{L_1 d}{2}\alpha_k h_k u_k^2.$$

By Assumption 4,

$$\mathbb{E}[u_{k+1}^2|\mathcal{F}_k] - u_k^2 \leq -2 \underbrace{\left(\frac{1}{L_1} - \frac{d}{\ell}\bar{\alpha}\right)}_{=:\Delta} \alpha_k \|\nabla f(x_k)\|^2 + \underbrace{\frac{L_1 d}{2}\alpha_k h_k}_{=:\rho_k} u_k^2$$

$$+ \underbrace{\frac{L_1 d}{2}\alpha_k h_k + \frac{L_1^2 d^2}{2\ell}\alpha_k^2 h_k^2}_{=:C_k}.$$

Note that $\Delta > 0$. Thus, rearranging the terms

$$\mathbb{E}[u_{k+1}^2|\mathcal{F}_k] - (1 + \rho_k)u_k^2 + 2\Delta\alpha_k\|\nabla f(x_k)\|^2 \leq C_k.$$

Since $\rho_k, C_k \in \ell^1$ by Assumption 4, Robbins-Siegmund Theorem [43] ensures that $(u_k^2)_{k\in\mathbb{N}}$ is convergent and $(\alpha_k\|\nabla f(x_k)\|^2)_{k\in\mathbb{N}} \in \ell^1$ a.s. Since $f$ is convex, it follows from (9) that

$$\mathbb{E}[u_{k+1}^2|\mathcal{F}_k] - (1 + \rho_k)u_k^2 \leq \frac{2d}{\ell}\alpha_k^2\|\nabla f(x_k)\|^2 - 2\alpha_k(f(x_k) - f(x^*)) + C_k.$$

Robbins-Siegmund Theorem [43] implies that $(\alpha_k(f(x_k) - f(x^*)))_{k\in\mathbb{N}} \in \ell^1$ a.s. Assumption 4 implies that $\alpha_k \notin \ell^1$ therefore

$$\liminf_k f(x_k) - f(x^*) = 0 \text{ a.s.} \tag{10}$$

By Lemma 8 and Assumption 4, we have that the sequence $\mathbb{E}[f(x_{k+1}) - f(x^*)|\mathcal{F}_k] - (f(x_k) - f(x^*))$ is upper-bounded by a sequence in $\ell^1$. Thus, by Robbins-Siegmund Theorem [43], $\lim_k(f(x_k) - f(x^*))$ exists a.s. Then, it follows from (10) that

$$\lim_{k\to\infty} f(x_k) = f(x^*) \quad a.s.$$

Moreover, as we saw before, $(\|x_k - x^*\|)_{k\in\mathbb{N}}$ is convergent a.s. for every $x^* \in \arg\min f$. Then, by Opial's Lemma [40], there exists a random variable $\hat{x}$ taking values in $\arg\min f$ such that $x_k \to \hat{x}$ a.s. $\qquad\square$

**Lemma 10** (Gradient bound: convex smooth setting). *Suppose that Assumptions 3 and 4 hold, and assume $f$ to be convex. Let $(x_k)_{k\in\mathbb{N}}$ be the sequence generated by Algorithm 1. Then, for every $k \in \mathbb{N}$ and every $x^* \in \arg\min f$,*

$$\sum_{i=0}^k \alpha_i \mathbb{E}[\|\nabla f(x_i)\|^2] \leq \frac{1}{2\Delta}\left(S_k + \sum_{i=0}^k \rho_i \sqrt{\mathbb{E}[\|x_i - x^*\|^2]}\right),$$

*and*

$$\sqrt{\mathbb{E}[\|x_k - x^*\|^2]} \leq \sqrt{S_{k-1}} + \sum_{i=0}^k \rho_i,$$

*where*

$$\Delta := \left(\frac{1}{L_1} - \frac{d}{\ell}\bar{\alpha}\right), \quad S_k := \|x_0 - x^*\| + \sum_{i=0}^k C_i$$

$$C_k := \frac{L_1^2 d^2}{2\ell}\alpha_k^2 h_k^2 \quad and \quad \rho_k := L_1 d\alpha_k h_k.$$

*Proof.* By Lemma 9 we derive

$$\mathbb{E}[\|x_{k+1} - x^*\|^2|\mathcal{F}_k] - \|x_k - x^*\|^2 \leq \frac{2d}{\ell}\alpha_k^2\|\nabla f(x_k)\|^2 + C_k$$
$$+ \rho_k\|x_k - x^*\| - 2\alpha_k\langle\nabla f(x_k), x_k - x^*\rangle.$$

By Baillon-Haddad Theorem and Assumption 4,

$$\mathbb{E}[\|x_{k+1} - x^*\|^2|\mathcal{F}_k] - \|x_k - x^*\|^2 \leq -2\Delta\alpha_k\|\nabla f(x_k)\|^2 + C_k + \rho_k\|x_k - x^*\|.$$

Let $u_k := \sqrt{\mathbb{E}[\|x_k - x^*\|^2]}$. Taking the full expectation, by Jensen inequality we have

$$u_{k+1}^2 - u_k^2 \leq -2\Delta\alpha_k \, \mathbb{E}[\|\nabla f(x_k)\|^2] + \rho_k u_k + C_k.$$

Summing the previous inequality from $i = 0, \cdots, k$, we get

$$u_{k+1}^2 + 2\Delta \sum_{i=0}^{k} \alpha_i \, \mathbb{E}[\|\nabla f(x_i)\|^2] \leq \underbrace{u_0^2 + \sum_{i=0}^{k} C_i}_{=:S_k} + \sum_{i=0}^{k} \rho_i u_i. \tag{11}$$

Since $u_k$ is non-negative, the first claim of the lemma follows. Since $\Delta > 0$, $\rho_k \geq 0$, $S_k$ is non decreasing, and $S_k \geq u_0^2$ in (11), then

$$u_{k+1}^2 \leq S_k + \sum_{i=0}^{k} \rho_i u_i.$$

Thus, the (discrete) Bihari's Lemma [32, Lemma 9.8] yields

$$u_{k+1} \leq \frac{1}{2} \sum_{i=0}^{k} \rho_i + \left[ S_k + \left( \frac{1}{2} \sum_{i=0}^{k} \rho_i \right)^2 \right]^{1/2} \leq \sqrt{S_k} + \sum_{i=0}^{k} \rho_i,$$

concluding the proof. $\qquad\square$

## B  Proofs of Main Results

### B.1  Proof of Theorem 1

By Lemma 6,

$$\mathbb{E}[\|x_{k+1} - x^*\|^2 | \mathcal{F}_k] - \|x_k - x^*\|^2 + 2\alpha_k(f(x_k) - f(x^*)) \leq 2c\frac{L_0^2 d}{\ell}\alpha_k^2 + 2L_0\alpha_k h_k.$$

Rearranging the terms, taking the full expectation, and summing the first $k$ iterations

$$\sum_{i=0}^{k} \alpha_i \, \mathbb{E}[(f(x_i) - f(x^*))] \leq \frac{\|x_0 - x^*\|^2}{2} + c\frac{dL_0^2}{\ell} \sum_{i=0}^{k} \alpha_i^2 + L_0 \sum_{i=0}^{k} \alpha_i h_i.$$

Let $\bar{x}_k := \sum_{i=0}^{k} \alpha_i x_i / (\sum_{i=0}^{k} \alpha_i)$. Dividing by $\sum_{i=0}^{k} \alpha_i$ and observing that by convexity we have

$$\mathbb{E}[f(\bar{x}_k) - \min f] \leq \frac{\sum_{i=0}^{k} \alpha_i \, \mathbb{E}[(f(x_i) - f(x^*))]}{\sum_{i=0}^{k} \alpha_i},$$

we get the first claim. Under Assumption 2, the second claim holds by Lemma 6.

### B.2  Proof of Corollary 1

By Theorem 1,

$$\mathbb{E}[f(\bar{x}_k) - \min f] \leq \frac{1}{\sum_{i=0}^{k} \alpha_i} \left( \frac{\|x_0 - x^*\|^2}{2} + c\frac{dL_0^2}{\ell} \sum_{i=0}^{k} \alpha_i^2 + L_0 \sum_{i=0}^{k} \alpha_i h_i \right).$$

Replacing $\alpha_k$ and $h_k$ with the sequences in the statement,

$$\mathbb{E}[f(\bar{x}_k) - f(x^*)] \leq \frac{C_1}{\alpha k^{1-\theta}} + \frac{C_2}{k^\rho} h + \frac{d}{\ell} \frac{C_3}{k^\theta} \alpha,$$

with

$$C_1 := \frac{(1-\theta)\|x_0 - x^*\|^2}{2}, \qquad C_2 := \frac{L_0(1-\theta)}{(1-\theta-\rho)} \quad \text{and} \quad C_3 := \frac{cL_0^2(1-\theta)}{(1-2\theta)}.$$

The second point of the corollary can be proved replacing $\alpha_k = \alpha$ and $h_k = h$. Now, to prove the third point, fix $\varepsilon \in (0,1)$. Since we want $\mathbb{E}[f(\bar{x}_k) - f(x^*)] \leq \varepsilon$, we impose

$$\frac{\|x_0 - x^*\|^2}{2\alpha k} + \frac{cdL_0^2}{\ell}\alpha + L_0 h \leq \varepsilon.$$

Choosing $h_k = h \leq \frac{\varepsilon}{2L_0}$, to get the previous inequality it is sufficient to impose

$$\frac{\|x_0 - x^*\|^2}{2\alpha k} + \frac{cdL_0^2}{\ell}\alpha \leq \frac{\varepsilon}{2}.$$

We fix a priori a number of iterations $K$ and we minimize the left handside with respect to $\alpha$, obtaining

$$\alpha = \sqrt{\frac{\ell}{d}} \frac{\|x_0 - x^*\|}{\sqrt{2cK}L_0}.$$

Thus, for $h_k = h \leq \frac{\varepsilon}{2L_0}$, $\alpha$ as above and

$$K \geq \frac{8\|x_0 - x^*\|^2 L_0^2 cd}{\ell\varepsilon^2},$$

we have $\mathbb{E}[f(\bar{x}_k) - f(x^*)] \leq \varepsilon$. Note that, since the computation of the surrogate requires $2\ell$ function evaluations, to ensure an error of $\varepsilon$ we need to perform a number of function evaluations of the order

$$\mathcal{O}(d\varepsilon^{-2}).$$

This concludes the proof.

### B.3 Proof of Theorem 2

By Lemma 7,

$$\mathbb{E}[f_h(x_{k+1})|\mathcal{F}_k] - f_h(x_k) \leq -\alpha_k\|\nabla f_h(x_k)\|^2 + c\frac{L_0^3 d\sqrt{d}}{\ell}\frac{\alpha_k^2}{h}.$$

Taking the full expectation and rearranging the terms,

$$\alpha_k \mathbb{E}[\|\nabla f_h(x_k)\|^2] \leq \mathbb{E}[f_h(x_k) - f_h(x_{k+1})] + c\frac{L_0^3 d\sqrt{d}}{\ell}\frac{\alpha_k^2}{h}.$$

Next sum from $i = 0$ to $i = k$. By definition of $f_h$, we have $f_h(x) \geq \min f$ for every $x \in \mathbb{R}^d$, thus,

$$\sum_{i=0}^{k} \alpha_i \mathbb{E}[\|\nabla f_h(x_i)\|^2] \leq \mathbb{E}[f_h(x_0) - \min f] + c\frac{L_0^3 d\sqrt{d}}{\ell}\sum_{i=0}^{k}\frac{\alpha_i^2}{h}. \tag{12}$$

The claim follows.

### B.4 Proof of Corollaries 2 and 3

By Theorem 2,

$$\eta_k^{(h)} \leq \left((f_h(x_0) - f(x^*)) + c\frac{L_0^3 d\sqrt{d}}{\ell}\sum_{i=0}^{k}\frac{\alpha_i^2}{h}\right)\Big/\left(\sum_{i=0}^{k}\alpha_i\right).$$

Due to the choice of $\alpha_k = \alpha(k+1)^{-\theta}$ with $\theta \in (1/2, 1)$ and $\alpha > 0$, we get

$$\eta_k^{(h)} \leq \frac{C_1}{\alpha(k+1)^{1-\theta}} + \frac{C_2 d\sqrt{d}\alpha}{\ell h}\frac{1}{(k+1)^\theta},$$

where

$$C_1 := \|x_0 - x^*\|^2 (1 - \theta) \quad \text{and} \quad C_2 := \frac{cL_0^3 (1 - \theta)}{(1 - 2\theta)}.$$

If we choose $\alpha_k = \alpha$, we derive

$$\eta_k^{(h)} \leq \frac{f_h(x_0) - \min f}{\alpha k} + \frac{cL_0^3 d\sqrt{d}\alpha}{\ell h}. \tag{13}$$

If we fix a priori a number of iteration $K$ and we minimize the right handside with respect to $\alpha$, we get

$$\hat{\alpha} = \sqrt{\frac{(f_h(x_0) - f(x^*))\ell h}{KcL_0^3 d\sqrt{d}}}.$$

Let $\varepsilon \in (0, 1)$. Choosing $\alpha = \hat{\alpha}$, we get $\eta_K^{(h)} \leq \varepsilon$ for

$$K \geq 4\frac{(f_h(x_0) - f(x^*))cL_0^3 d\sqrt{d}}{\ell h} \varepsilon^{-2}. \tag{14}$$

This concludes the proof of Corollary 2. To prove Corollary 3, we fix a maximum number of iterations $K \in \mathbb{N}$ and consider the random variable $I$ of the statement. Let $\partial_h f$ be the $h$-Goldstein subdifferential defined in Definition 1. It follows from [34, Theorem 3.1] that $\nabla f_h(x_I) \in \partial_h f(x_I)$ almost surely, therefore

$$\mathbb{E}_I \min[\|\eta\|^2 \ : \ \eta \in \partial_h f(x_I)] \leq \mathbb{E}_I \mathbb{E}[\|\nabla f_h(x_I)\|^2].$$

In addition, Theorem 2 yields

$$\begin{aligned}
\mathbb{E}_I \mathbb{E}_G[\|\nabla f_h(x_I)\|^2] &= \left( \sum_{j=0}^{K-1} \alpha_j \mathbb{E}_G[\|\nabla f_h(x_j)\|^2] \right) / \sum_{j=0}^{K-1} \alpha_j \\
&\leq \mathbb{E}[f_h(x_0) - \min f] + c\frac{L_0^3 d\sqrt{d}}{\ell} \sum_{i=0}^{k} \frac{\alpha_i^2}{h}.
\end{aligned}$$

Thus,

$$\mathbb{E}_I[\|\eta\|^2 \ : \ \eta \in \partial_h f(x_I)] \leq \mathbb{E}_I \mathbb{E}[\|\nabla f_h(x_I)\|^2] = \eta_k^{(h)}.$$

Hence, for $\alpha = \bar{\alpha}$ and $K$ chosen s.t. inequality (14) holds, we have

$$\mathbb{E}_I[\|\eta\|^2 \ : \ \eta \in \partial_h f(x_I)] \leq \varepsilon.$$

This concludes the proof.

## B.5 Proof of Theorem 3

By Lemma 9,

$$\begin{aligned}
\mathbb{E}[\|x_{k+1} - x^*\|^2 | \mathcal{F}_k] - \|x_k - x^*\|^2 &\leq \frac{2d}{\ell}\alpha_k^2 \|\nabla f(x_k)\|^2 + 2\alpha_k \langle \nabla f(x_k), x^* - x_k \rangle \\
&\quad + \underbrace{L_1 d\alpha_k h_k}_{=:\rho_k} \|x^* - x_k\| + \underbrace{\frac{L_1^2 d^2}{2\ell}\alpha_k^2 h_k^2}_{=:C_k}.
\end{aligned}$$

By convexity,

$$\begin{aligned}
\mathbb{E}[\|x_{k+1} - x^*\|^2 | \mathcal{F}_k] - \|x_k - x^*\|^2 &\leq \frac{2d}{\ell}\alpha_k^2 \|\nabla f(x_k)\|^2 - 2\alpha_k (f(x_k) - f(x^*)) \\
&\quad + \rho_k \|x^* - x_k\| + C_k.
\end{aligned}$$

Rearranging the terms and taking the full expectation,

$$\begin{aligned}
2\mathbb{E}[\alpha_k (f(x_k) - f(x^*))] &\leq \mathbb{E}[\|x_k - x^*\|^2 - \|x_{k+1} - x^*\|^2] + \frac{2d}{\ell}\alpha_k^2 \mathbb{E}[\|\nabla f(x_k)\|^2] \\
&\quad + \rho_k \mathbb{E}[\|x^* - x_k\|] + C_k.
\end{aligned}$$

Since $\mathbb{E}[\|x^* - x_k\|] = \mathbb{E}[\sqrt{\|x^* - x_k\|^2}]$, Jensen's inequality implies that

$$2\,\mathbb{E}[\alpha_k(f(x_k) - f(x^*))] \leq \mathbb{E}[\|x_k - x^*\|^2 - \|x_{k+1} - x^*\|^2] + \frac{2d}{\ell}\alpha_k^2\,\mathbb{E}[\|\nabla f(x_k)\|^2]$$
$$+ \rho_k\sqrt{\mathbb{E}[\|x^* - x_k\|^2]} + C_k.$$

Denoting with $u_k = \mathbb{E}[\|x_k - x^*\|^2]$ and taking the sum from $i = 0$ to $i = k$,

$$2\sum_{i=0}^{k}\alpha_i\,\mathbb{E}[f(x_i) - f(x^*)] \leq \underbrace{u_0^2 + \sum_{i=0}^{k}C_i}_{=:S_k} + \frac{2d}{\ell}\sum_{i=0}^{k}\alpha_i^2\,\mathbb{E}[\|\nabla f(x_i)\|^2] + \sum_{i=0}^{k}\rho_i u_i$$

$$\leq S_k + \frac{2d}{\ell}\bar{\alpha}\sum_{i=0}^{k}\alpha_i\,\mathbb{E}[\|\nabla f(x_i)\|^2] + \sum_{i=0}^{k}\rho_i u_i,$$

where the last inequality holds by Assumption 4. Let $\Delta := (1/L_1 - (d/\ell)\bar{\alpha})$. By Lemma 10, we have

$$\sum_{i=0}^{k}\alpha_i\,\mathbb{E}[f(x_i) - f(x^*)] \leq \frac{1}{2}\left(S_k + \frac{d\bar{\alpha}}{\Delta\ell}\left[S_k + \sum_{i=0}^{k}\rho_i u_i\right] + \sum_{i=0}^{k}\rho_i u_i\right)$$

$$= \frac{\ell\Delta + d\bar{\alpha}}{2\ell\Delta}\left(S_k + \sum_{i=0}^{k}\rho_i u_i\right)$$

$$\leq \frac{\ell\Delta + d\bar{\alpha}}{2\ell\Delta}\left(S_k + \sum_{i=0}^{k}\rho_i(\sqrt{S_i} + \sum_{j=0}^{i}\rho_j)\right).$$

Let $\bar{x}_k := \sum_{i=0}^{k}\alpha_i x_i / (\sum_{i=0}^{k}\alpha_i)$. Dividing both sides by $\sum_{i=0}^{k}\alpha_i$, convexity yields

$$\mathbb{E}[f(\bar{x}_k) - \min f] \leq \frac{\displaystyle\sum_{i=0}^{k}\alpha_i\,\mathbb{E}[(f(x_i) - f(x^*))]}{\displaystyle\sum_{i=0}^{k}\alpha_i}.$$

## B.6   Proof of Corollary 4

In this proof, we use the same notation as the one in the proof of Theorem 3. By the choices of the parameters, we have

$$\sum_{i=0}^{k}\rho_i \leq C_1 d\alpha h \quad\text{with}\quad C_1 := \frac{L_1\theta}{\theta - 1},$$

$$S_k \leq \|x_0 - x^*\|^2 + C_2\frac{d^2}{\ell}\alpha^2 h^2 \quad\text{with}\quad C_2 := \frac{L_1^2\theta}{2\theta - 1}.$$

Thus, using these inequalities in Theorem 3, we get

$$D_k \leq \frac{\ell\Delta + d\bar{\alpha}}{2\ell\Delta}\left(\|x_0 - x^*\|^2 + C_2\frac{d^2\alpha^2 h^2}{\ell} + \sqrt{\|x_0 - x^*\|^2}C_3 d\alpha h\right.$$
$$\left. + C_4\frac{d\alpha h}{\sqrt{\ell}} + C_5 d^2\alpha^2 h^2\right),$$

with

$$C_3 := \frac{L_1^2\theta}{\theta - 1}, \quad C_4 := \frac{L_1^2}{\sqrt{2}}, \quad C_5 := \frac{L_1^2\theta}{(\theta - 1)^2}.$$

Dividing by $\sum_{i=0}^{k}\alpha_i$, we get

$$\mathbb{E}[f(\bar{x}_k) - \min f] \leq \frac{C}{\alpha k}.$$

Note that by Assumption 4, $\alpha < \ell/(dL_1)$, thus $1/\alpha > (dL_1)/\ell$. The algorithm performs $2\ell$ function evaluations at each iteration. Thus, to guarantee $\mathbb{E}[f(\bar{x}_k) - \min f] \leq \varepsilon$ for $\varepsilon \in (0,1)$, the algorithm has to perform a number of function evaluations in the order of

$$\mathcal{O}(d\varepsilon^{-1}).$$

Assuming, instead, $\alpha_k \leq \bar{\alpha} < \ell/(2dL_1)$, by Lemma 9 we get the last claim; i.e, there exists a random variable $\hat{x}$ taking values in $\arg \min f$ s.t. $x_k \to \hat{x}$ a.s.

### B.7 Proof of Theorem 4

Set $C_1 = (dL_1)/2$. It follows from Lemma 8 that

$$\mathbb{E}[f(x_{k+1})|\mathcal{F}_k] - f(x_k) \leq -\left(\frac{1}{2} - \frac{L_1 d}{\ell}\bar{\alpha}\right)\alpha_k\|\nabla f(x_k)\|^2 + \frac{C_1^2 \alpha_k h_k^2}{2} + \frac{L_1^3 d^2}{4\ell}\alpha_k^2 h_k^2.$$

Taking the full expectation and rearranging the terms, and recalling the definition of $\Delta$,

$$\Delta \alpha_k \, \mathbb{E}[\|\nabla f(x_k)\|^2] \leq \mathbb{E}[f(x_k) - f(x_{k+1})] + \frac{C_1^2 \alpha_k h_k^2}{2} + \frac{L_1^3 d^2}{4\ell}\alpha_k^2 h_k^2.$$

Summing for $i = 0, \cdots, k$ and observing that $\min f \leq f(x)$ for every $x$,

$$\Delta \sum_{i=0}^{k} \alpha_i \, \mathbb{E}[\|\nabla f(x_i)\|^2] \leq f(x_0) - \min f + \sum_{i=0}^{k} \frac{C_1^2 \alpha_i h_i^2}{2} + \frac{L_1^3 d^2}{4\ell} \sum_{i=0}^{k} \alpha_i^2 h_i^2.$$

Dividing by $\Delta \sum_{i=0}^{k} \alpha_i$ we get the claim.

### B.8 Proof of Corollary 5

$(i)$: From the choice of $\alpha_k$ and $h_k$, we have

$$\sum_{i=0}^{k} \alpha_i h_i^2 \leq \frac{2\theta \alpha h^2}{2\theta - 1} \qquad \sum_{i=0}^{k} \alpha_i^2 h_i^2 \leq \frac{2\theta \alpha^2 h^2}{2\theta - 1}.$$

It follows from Theorem 4 that

$$\eta_k \leq \frac{1}{\Delta \alpha k}\left(f(x_0) - \min f + C_1 d^2 \alpha h^2 + \frac{C_2 \alpha^2 h^2 d^2}{\ell}\right),$$

with $C_1 = \frac{L_1^2 \theta}{4(2\theta - 1)}$ and $C_2 = \frac{L_1^3 \theta}{2(2\theta - 1)}$.
$(ii)$: It follows directly from Theorem 4 taking into account that

$$\sum_{i=0}^{k} \alpha_i h_i^2 = k\alpha h^2, \qquad \sum_{i=0}^{k} \alpha_i^2 h_i^2 = k\alpha^2 h^2,$$

and setting $C_1 = L_1^2/8$ and $C_2 = L_1^3/4$.

## C  Experimental Details

In this appendix, we report details on the experiments performed. We implemented every script in Python3 (version 3.9.11) and used numpy (version 1.22.2) [27] and matplotlib (version 3.5.1) [29] libraries.

**Machine used to perform the experiments.**  In the following table, we describe the features of the machine used to perform the experiments in Section 4.

Table 1: Machine used to perform the experiments

| Feature | |
| --- | --- |
| OS | Debian GNU/Linux 11 |
| CPU(s) | 4 x Intel(R) Core(TM) i7-1165G7 11th Gen @ 2.80GHz |
| CPU Core(s) | 4 |
| RAM | 8 GB |

**Target Functions.** We considered two synthetic target functions: a convex smooth function $f_1$ and a convex non-smooth function $f_2$ defined as follows

$$\text{(Convex Smooth)} \quad f_1(x) := \frac{1}{2}\|Ax\|^2 \quad \text{with} \quad A \in \mathbb{R}^{d \times d}$$

$$\text{(Convex Non-smooth)} \quad f_2(x) := \|x - \bar{v}\|_1$$

where $A$ is a random Gaussian matrix (i.e. $A_{i,j} \sim \mathcal{N}(0,1)$) and $\bar{v} := [0, 1, \cdots, d-1]^\intercal$.

**Choice of the number of directions.** We report here the details of the first experiment of Section 4. For these experiments, we consider $d = 50$ and we use, for the smooth convex case, the following parameters

$$\alpha_k = 0.99 \frac{\ell}{dL_1} \quad \text{and} \quad h_k = \frac{10^{-5}}{k+1}.$$

The constant $L_1$ is computed as the maximum eigenvalue of the matrix $A^\intercal A$. Note that this parameter choice satisfies Assumption 4. For the non-smooth target, we used

$$\alpha_k = \sqrt{\frac{\ell}{d}} k^{-1/2 - 10^{-5}} \quad \text{and} \quad h_k = \frac{10^{-7}}{k+1}.$$

Note that this parameter configuration satisfies Assumption 2. The maximum number of function evaluations considered is $4000$. The direction matrices $G_k$ are generated with the QR method - see Appendix D.

**Comparison with Finite-difference methods.** In Section 4, we compare finite-difference method with different choice of directions. In order to make a fair comparison we consider only central finite-differences. However, note that Algorithm 1 can be modified (in practice) considering computationally cheaper gradient estimators - see Remark 1. For these experiments, we consider $d = 10$ and $\ell = d$ for methods with multiple directions. The maximum number of function evaluations is $1000$ for both smooth and non-smooth targets and the direction matrices $G_k$ for Algorithm 1 are generated with the QR method - see Appendix D. To solve the smooth problem we consider the following parameter choice for every method

$$\alpha_k = c \frac{\ell}{dL_1} \quad \text{and} \quad h_k = \frac{10^{-7}}{d^2(k+1)},$$

where $L_1$ is computed taking the maximum eigenvalue of $A^\intercal A$. For Algorithm 1 and finite-difference with single and multiple spherical directions $c = 0.99$ while it is equal to $c = 0.11$ for finite-difference with single and multiple Gaussian directions. We made this choice since for finite-difference methods with Gaussian directions we observed divergence for larger choices of $c$ - see Figure 3.

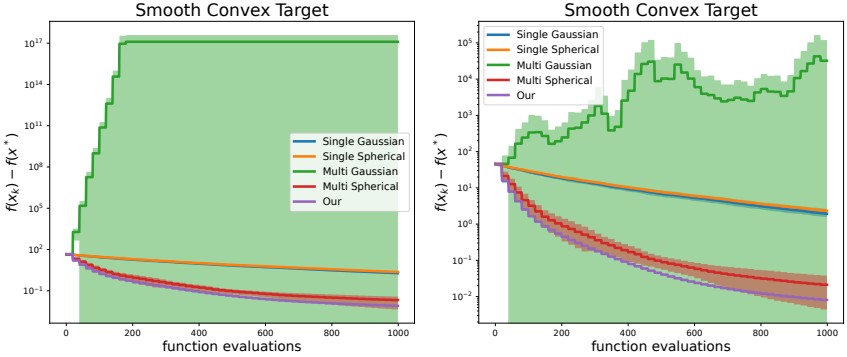

Figure 3: From left to right, comparison of finite-difference methods for smooth convex target with $c = 0.99$ and $c = 0.2$ for methods with Gaussian directions.

For the non-smooth convex target, we considered the following parameter choice

$$\alpha_k = c \frac{\ell}{d} k^{-1/2 - 10^{-5}} \quad \text{and} \quad h_k = \frac{1}{d^2(k+1)}.$$

For every method, we selected $c = 0.65$ except for the method with multiple Gaussian directions in which we selected $c = 0.08$ since it provided better performances - see Figure 4.

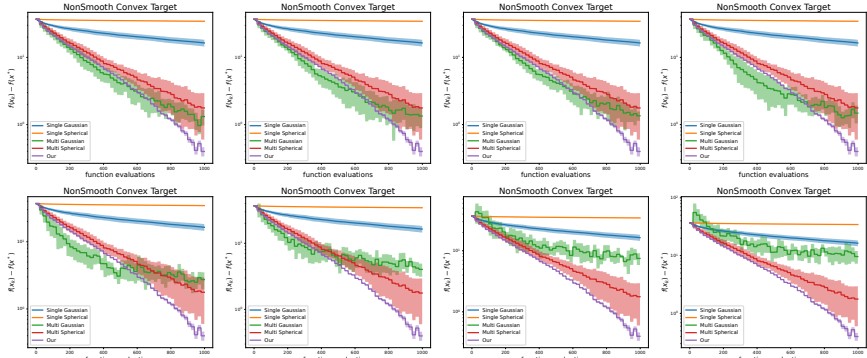

Figure 4: From left to right and up to down, comparison of finite difference method with different directions and different values of $c$ for multiple Gaussian directions. The values of $c$ considered are the following $[0.085, 0.089, 0.09, 0.1, 0.2, 0.3, 0.5, 0.65]$

# D    Techniques to Generate Orthogonal Direction Matrices

In the literature, different algorithms were proposed to generate orthogonal matrices - see for instance [23, 38, 11, 28, 7, 2, 4, 44, 8] and references therein. Such methods can be used to generate the direction matrices $G_k$ required for the iteration proposed in Algorithm (1). In this appendix, we briefly discuss three of them.

**QR factorization.** As observed in [32, 42], a way to generate orthogonal consists in generating a random Gaussian matrix $A \in \mathbb{R}^{d \times d}$ with $A_{i,j} \sim \mathcal{N}(0, 1)$ and perform the QR factorization i.e. $A = QR$. Then, the direction matrix is the truncation of the $Q$ matrix i.e. $QI_{d,\ell}$.

**Householder Reflection.** To obtain a direction matrix, we can use a Householder reflector. This can be done by sampling a vector $v$ from the unit sphere $\mathbb{S}^{d-1}$. The direction matrix $G$ is defined as a Householder reflector, given by

$$G := I - 2vv^{\mathsf{T}},$$

with $I \in \mathbb{R}^{d \times d}$ identity matrix. To obtain the desired matrix, we compute the product of $G$ with $I_{d,\ell}$, i.e., we take the first $\ell$ columns. The (truncated) identity matrix can be generated and stored offline (note that since it is very sparse, it can be stored using a sparse format (e.g. the COO format proposed in scikit-learn library[9]). In this way, we can save resources in high-dimensional settings. In order to quantify the time-cost of this procedure, we compared the time of generating this kind of matrix with random matrices with different dimensions. For this experiment, we consider the $\ell = d$ case i.e. the most expensive. Matrices are computed in CPU and the details of the machine used are described in Appendix C. We report the mean and standard deviation of the time using 500 repetitions. In Figure 5, we compare the time-cost of generating orthogonal matrices with this procedure against generating random matrices while in Table 2 we report the mean and standard deviation of the results.

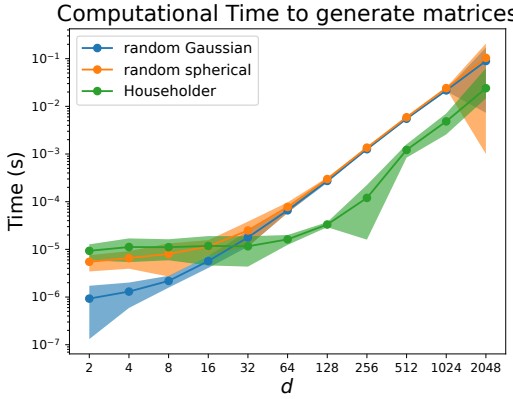

Figure 5: Time comparison in CPU of different methods to generate direction matrices.

In Figure 5, we can observe that using this strategy we can limit the cost of generating random orthogonal matrices. In particular, for dimensions larger than 32, our method is faster than random gaussian and spherical directions.

Table 2: Comparison of the time-cost (seconds) of generating random and orthogonal matrices with different dimensions

| $d$ | Random Gaussian | Random Spherical | Householder |
|---|---|---|---|
| 2 | $9.27 \times 10^{-7} \pm 7.96 \times 10^{-7}$ | $5.49 \times 10^{-6} \pm 2.05 \times 10^{-6}$ | $9.32 \times 10^{-6} \pm 3.34 \times 10^{-6}$ |
| 4 | $1.30 \times 10^{-6} \pm 7.21 \times 10^{-7}$ | $6.56 \times 10^{-6} \pm 2.63 \times 10^{-6}$ | $1.12 \times 10^{-5} \pm 5.79 \times 10^{-6}$ |
| 8 | $2.18 \times 10^{-6} \pm 6.06 \times 10^{-7}$ | $8.01 \times 10^{-6} \pm 5.32 \times 10^{-6}$ | $1.11 \times 10^{-5} \pm 5.20 \times 10^{-6}$ |
| 8 | $2.18 \times 10^{-6} \pm 6.06 \times 10^{-7}$ | $8.01 \times 10^{-6} \pm 5.32 \times 10^{-6}$ | $1.11 \times 10^{-5} \pm 5.20 \times 10^{-6}$ |
| 16 | $5.69 \times 10^{-6} \pm 1.61 \times 10^{-6}$ | $1.15 \times 10^{-5} \pm 4.10 \times 10^{-6}$ | $1.18 \times 10^{-5} \pm 7.20 \times 10^{-6}$ |
| 32 | $1.78 \times 10^{-5} \pm 6.42 \times 10^{-6}$ | $2.49 \times 10^{-5} \pm 1.33 \times 10^{-5}$ | $1.16 \times 10^{-5} \pm 7.25 \times 10^{-6}$ |
| 64 | $6.58 \times 10^{-5} \pm 7.03 \times 10^{-6}$ | $7.74 \times 10^{-5} \pm 1.95 \times 10^{-5}$ | $1.62 \times 10^{-5} \pm 3.79 \times 10^{-6}$ |
| 128 | $2.73 \times 10^{-4} \pm 2.37 \times 10^{-5}$ | $2.98 \times 10^{-4} \pm 2.45 \times 10^{-5}$ | $3.32 \times 10^{-5} \pm 4.02 \times 10^{-6}$ |
| 256 | $1.26 \times 10^{-3} \pm 2.79 \times 10^{-5}$ | $1.36 \times 10^{-3} \pm 2.90 \times 10^{-5}$ | $1.20 \times 10^{-4} \pm 1.04 \times 10^{-4}$ |
| 512 | $5.50 \times 10^{-3} \pm 1.63 \times 10^{-4}$ | $5.91 \times 10^{-3} \pm 1.22 \times 10^{-4}$ | $1.22 \times 10^{-3} \pm 3.82 \times 10^{-4}$ |
| 1024 | $2.16 \times 10^{-2} \pm 6.92 \times 10^{-4}$ | $2.41 \times 10^{-2} \pm 7.35 \times 10^{-4}$ | $4.83 \times 10^{-3} \pm 2.26 \times 10^{-3}$ |
| 2048 | $8.92 \times 10^{-2} \pm 8.19 \times 10^{-2}$ | $1.04 \times 10^{-1} \pm 1.03 \times 10^{-1}$ | $2.40 \times 10^{-2} \pm 3.87 \times 10^{-2}$ |

Moreover, if more computational resources are available, we can build $m > 1$ Householder reflectors $G_1, \cdots, G_m$ using $m$ random vectors $v_1, \cdots, v_m$ sampled i.i.d from $\mathbb{S}^{d-1}$ and define the direction matrix as

$$G_1 G_2 \cdots G_m I_{d,\ell}.$$

It is important to note that when $m = d$, this procedure is equivalent to using the QR factorization.

**Haar Butterfly matrices.** We can build orthogonal matrices using Butterfly matrices [48]. Let $G^{(0)} := [1]$, we can build an orthogonal matrix of dimension $d = 2^n$ with the following recursion

$$G^{(n)} = \begin{bmatrix} \cos(\theta_n) G^{(n-1)} & \sin(\theta_n) G^{(n-1)} \\ -\sin(\theta_n) G^{(n-1)} & \cos(\theta_n) G^{(n-1)} \end{bmatrix}$$

where $\theta_n$ is sampled uniformly in $[0, 2\pi]$. Then we compute $G I_{d,\ell}$ (we take the first $\ell$ columns). The construction of Haar butterfly matrices is faster than previous methods because it only requires simple operations. However, this procedure allows to build only matrices with $d = 2^n$ for $n \geq 0$. In literature, different methods were proposed to cope with this limitation e.g. [23].

# E Limitations

In this appendix, we discuss the main practical limitations of Algorithm 1. Like all finite-difference methods with multiple directions, O-ZD requires multiple function evaluations to execute a single step. In many practical applications, function evaluations can be time-consuming, leading to the use of a small number of directions $\ell$. This may result in poor performance as observed in numerical experiments. As for the subgradient method, in O-ZD the step size significantly affects performance, and tuning it can be challenging. To address this limitation, an adaptive stepsize selection method could be proposed. Furthermore, decreasing the sequence $h_k$ too quickly can lead to numerical instability, as noted in [42].

# F Other Experiments

We performed other experiments in minimizing convex functions. We considered the targets defined in Table 3 and, for each experiment, we reported the mean and standard deviation using 20 repetitions.

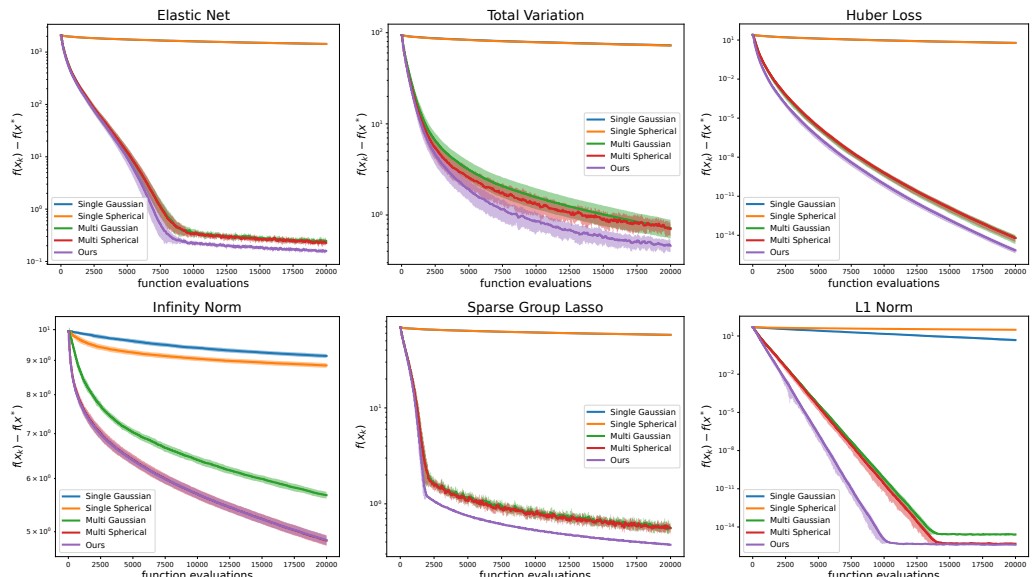

Figure 6: Function values per function evaluation in optimizing functions with different algorithms.

In Figure 6, we can observe that structured finite-difference performs better than unstructured methods.

Table 3: Functions used and relative dimension and number of directions considered.

| Name | Definition | $d$ | $\ell$ |
|---|---|---|---|
| Sparse Group Lasso | $f(x) := \sum\limits_{i=1}^{p} \|x^{(\beta_i)}\|$ | 50 | 25 |
| Huber Loss | $f(x) := \begin{cases} 0.5\|x\|_2^2 & \|x\|_2 \leq \delta \\ \delta\|x\|_2 - 0.5\delta^2 & \text{otherwise} \end{cases}$ for $\delta > 0$ | 50 | 25 |
| Elastic Net | $f(x) := \alpha\|x\|_1 + 0.5\beta\|x\|_2^2$ | 50 | 25 |
| L1 | $f(x) := \|x\|_1$ | 50 | 25 |
| Infinity Norm | $f(x) := \|x\|_\infty$ | 50 | 20 |
| Total Variation | $f(x) := \|x\|_{\text{TV}}$ | 50 | 25 |

In Table 3, we define the function used for the experiments. In particular:

- Sparse Group Lasso: $p$ is set to 3 and given an $x \in \mathbb{R}^d$, $x^{(\beta_i)}$ is a vector obtained by taking 3 entries of $x$.
- Huber Loss: $\delta$ is set to $0.5$.
- Elastic Net: $\alpha, \beta$ are set to $0.5$.

