# OpenReview forum: "An Optimal Structured Zeroth-order Algorithm for Non-smooth Optimization"
_NeurIPS.cc/2023/Conference — NeurIPS 2023 poster_

### Official Review · Reviewer_RtjB · 2023-06-17

**Soundness:** 3 good
**Presentation:** 3 good
**Contribution:** 2 fair
**Rating:** 5
**Confidence:** 3

**Summary:**

This paper presents a study on a structured finite-difference algorithm for non-smooth black box optimization. The authors successfully demonstrate that their finite-difference surrogate serves as an unbiased estimator of the gradient for the smoothing approximation of the target function. The proposed O-ZD method's convergence analysis is established under different assumptions.

**Strengths:**

Strengths And Weaknesses:
Overall, the paper is well-structured and provides valuable insights into the structured finite-difference method for non-smooth black box optimization. The authors establish the optimal complexity in the non-smooth convex case and demonstrate convergence rates in the non-smooth nonconvex and smoothing settings. However, it is unclear to me what is the exact theoretical complexity improvement of this method compared to state-of-the-art methods in different scenarios. Additionally, the objective function used in the numerical experiments appears to be relatively simple.

Furthermore, I kindly request the authors to address the following questions for better understanding:

1. Could the authors please provide the exact formula of C in the Corollary 1? It seems that C should be somehow related to \theta.

2. In line 205, the authors claim that the complexity in terms of the number of iterations is better than [15] and [37]. However, I believe it would be more appropriate to compare the number of function evaluations since the iterations of O-ZD are more computationally expensive.

3. I suggest the authors include a table explicitly listing the number of function evaluations' complexity of this work and the related works, rather than just providing a discussion for each case separately.

**Weaknesses:**

See above

**Questions:**

See above

---

> ### Author Rebuttal · Authors · 2023-08-07
>
> We thank the reviewer for his/her observations and questions. We answer here to his/her questions.
>
> ## Weakness
> In the literature, different works (empirically) showed that imposing a structure on the direction matrices let zeroth order method obtain better performance. However, no non-smooth analysis was provided. The main goal of this work was to provide analysis of structured zeroth order method in non-smooth setting. Indeed, one of our main results is the Smoothing Lemma for direction matrices in $\mathcal{O}(d)$. In terms of advantages w.r.t. the state-of-the-art is in the variance of the gradient estimator (as indicated in lines 210 to 212). Moreover, we agree with the reviewer that the objective functions considered in the numerical experiments are relatively simple. Indeed, the main contributions of this work are theoretical and the experimental part is included just to confirm the theoretical results. We will extend the experimental part by repeating the experiments on different functions (some of these experiments are included in the global response).
>
> ## Questions
> * Q1: We provided the explicit formulation of the constant $C$ of Corollary $1$ (point $(i)$) in the proof (in Appendix B.2). We will write it in the main paper.
> * Q2 and Q3: In line 205 we discuss the result obtained for non-smooth convex functions. In [30] and [15, Theorem 2], the authors consider estimators built using a single directions and thus the complexity in terms of number of iterations coincide with the complexity in terms of function evaluations (i.e. we can obtain such a complexity by multiplying the complexity in terms of number of iterations by $2$ since a single iteration costs $2$ function evaluations). In [30] (more precisely in Theorem 6), to derive the complexity, authors consider a constant step-size scheme (see [30] eq. 46) and they obtain a complexity of $\mathcal{O}(d^2 \varepsilon^{-2})$. For our algorithm, the constant step-size scheme is considered in Corollary $1$ point $(iii)$. The complexity in terms of function evaluations is $\mathcal{O}(d \varepsilon^{-2})$ (since every iteration costs $2\ell$ function evaluations, we can compute the complexity in terms of number of function evaluations by multiplying the complexity in terms of number of iterations by $2\ell$). In [15, Theorem 2], the authors consider a stepsize sequence $\alpha_k$ that satisfies the Monroe conditions. More precisely, they consider a stepsize sequence $\alpha_k$ as $\alpha (1/\sqrt{k})$ with $\alpha$ constant (note that it does not satisfy the Monroe conditions, however the choice $1/(k^{1/2 + \delta})$ satisfies it with $\delta$ arbitrary near to $0$). Using their double smoothing scheme they obtain a complexity in terms of function evaluations in the order of $\mathcal{O}(d \log d \varepsilon^{-2})$ (you can obtain it upper-bounding eq. $18$ by $\varepsilon$ and solving the inequality for $k$ i.e. searching $k$ s.t. the right part of eq. $18$ is smaller than $\varepsilon$). To make a fair comparison, we consider the choice of parameters proposed in Corollary 1 point $(i)$ i.e. $\alpha_k = \alpha k^{-\theta}$. As indicated in the discussion, in order to obtain the optimal dependence on the dimension we need to include $\sqrt{\ell/d}$ in the stepsize e.g. by taking $\alpha = \sqrt{\ell/d}$ (line 204-205). Again the complexity in terms of function evaluations is $\mathcal{O}(d \varepsilon^{-2})$.
> We will include an Appendix "Expanded Discussion" in which we extend the discussions below the corollaries including also the table with complexities in terms of the number of function evaluations of this work and the related works.

---

> > ### Comment · Reviewer_RtjB · 2023-08-16
> >
> > Thank you for addressing my concerns. I have raised my score.

---

### Official Review · Reviewer_77xV · 2023-06-27

**Soundness:** 3 good
**Presentation:** 3 good
**Contribution:** 3 good
**Rating:** 7
**Confidence:** 3

**Summary:**

This paper analyzes the convergence rate of structured zeroth order optimization, Whose iterations’ descent direction Is chosen by random orthogonal group. It applies to the most general non smooth setting, and this paper gives the convergence rate of all specific settings of interest.

**Strengths:**

This paper analyzes an important optimization method, and provides concrete mathematical analysis to prove the theories. The result is novel and comprehensive. This paper has a very detailed discussion of many settings, each of them has the convergence rate in its case.

**Weaknesses:**

No major weaknesses, questions below.

**Questions:**

In Eq(2), is there a motivation why you want to fix $l$ and vary $G$, is $h$ fixed or random?

In Algo. 1, I think it means sample $G_k$ i.i.d. from “a uniform distribution” on $O(d)$, is that correct?

What structure do the Algo’s chosen direction satisfy? For example, the paper mentions “orthogonality [30,40]” as a type of structure, but are the directions in Algo. 1 orthogonal? What is the difference between 1) “sampling $G_k$ randomly” and 2) zeroth order GD when you just randomly choose a point nearby and estimate $f(x+dx) – f(x-dx)$? With this part clarified, I would be willing to increase the score.

There is another type of zeroth order method – although not always applicable in practice but worth mention. When the objective function is analytic, one can use complex number to make estimation variance smaller by estimating the gradient by
$$ Im( f(x+yi) – f(x-yi) ) / 2y $$
where $y$ is small. For example, if $f(x) = x^3$, then
$$ Im( f(x+yi) – f(x-yi) ) / 2y = 3x^2 + O(y^2)$$
$$ ( f(x+y) – f(x-y) ) / 2y = 3x^2 + O(y)$$
With complex number, the noise or variance on the real part can be ignored, with only high order noise kept.

======

Raised to 7.

---

> ### Author Rebuttal · Authors · 2023-08-07
>
> We thank the reviewer for his/her comments. We answer here to his/her questions.
>
> ## Question
>
> **Eq. 2 and gradient estimator.** In eq.2, we introduce our gradient estimator. The parameter $h$ controls the smoothness of the smoothed target (see Proposition 1) and, in the smooth setting, the quality of the estimator (see Lemma 4). It is not random, but it is fixed (and the results depend on the choice of this parameter). Specifically, in Eq. 2 $h$ is fixed. In Algorithm 1 we consider a sequence of $h_k$ where $k$ is the iteration counter, and for every $k=0, 1, 2, \cdots$,  we compute
> \begin{equation}
> 	g_k(x_k) := \frac{d}{\ell} \sum\limits\_{i = 1}^\ell \frac{f(x_k + h_k G_k e_i) - f(x_k - h_k G_k e_i)}{2h_k} G_k e_i.
> \end{equation}
> According to the theoretical result, the best choice of $\ell$  is $\ell =d$ because it let reduce or remove (in non-smooth setting) the dependence on the dimension in the variance upper bound (see Lemma 4 in Appendix). However, a sequence of $\ell$ can be considered and it can be useful in pratical scenarios where a budget of function evaluations is provided (and it can be "derived" by considering the cost in time of the single function evaluation and how much time we want to spend to solve the optimization problem).
>
> **$G_k$ sampling.** In Algorithm 1, $G_k$ is i.i.d. uniformly sampled from $O(d)$.
>
> **Structured directions and differences with random directions.** Directions in Algorithm 1 are orthogonal in the sense that, since $G_k \in \mathcal{O}(d)$, it implies that $G_k^\intercal G_k = G_k G_k^\intercal = I$. Of course, since $\ell \leq d$ we have to truncate it. The main difference between using orthogonal directions and random directions is in the variance of the estimator obtained as we indicated in line 210-212. Moreover, in previous works (see e.g. [1, 2, 3, 4]), authors showed that gradient estimators built  with structured directions (in particular orthogonal directions) provide better performance than the ones built using random directions. In particular, in [2] the authors empirically show that to obtain a gradient accuracy comparable to methods that use orthogonal directions, other methods (e.g. random Gaussian or spherical) can require significantly more samples. Informally, such an improvement can be justified by noting that structured directions provide a better local exploration of the space than non-structured ones, reducing the probability to generate bad directions.
>
> **Other zeroth-order method.** We thank the reviewer to indicate these zeroth-order methods. We will include some references in the "Related Work" section.
>
> **References**
> 1. K. Choromanski, M. Rowland, V. Sindhwani, R. Turner, and A. Weller. Structured evolution with compact architectures for scalable policy optimization.
> 2. A. S. Berahas, L. Cao, K. Choromanski, and K. Scheinberg. A theoretical and empirical comparison of gradient approximations in derivative-free optimization.
> 3. M. Rando, C. Molinari, S. Villa, and L. Rosasco. Stochastic zeroth order descent with structured directions.
> 4. D. Kozak, C. Molinari, L. Rosasco, L. Tenorio, and S. Villa. Zeroth order optimization with orthogonal random directions

---

> > ### Comment · Reviewer_77xV · 2023-08-12
> > **Thanks for the comment**
> >
> > The comment mostly makes change to me.
> >
> > About the "structure" part, I see that it means in each batch, since $e_i$'s are orthorgonal, then "in each batch" the directions are orthogonal. Could the authors point how much faster it is compared to just sampling random vectors (given Reviewer KbSU saying that sampling $G$ might have higher oracle complexity)? Raised score to 6 for now, will make it 7 if the authors reply a convincing answer to this and to Reviewer KbSU's question.
> >
> > About fixed $l$, it would be great to find an optimal rate with the best strategy of "a sequence of $l$ can be considered".
> >
> > I think the theory is self-consistent and no more experiments are necessary in opposition to Reviewer VJ1k's comment.

---

> > > ### Author Response · Authors · 2023-08-14
> > >
> > > We thank the reviewer for his/her response. We answer here to his/her question.
> > > Note that in the literature, different methods to generate orthogonal matrices were proposed - see e.g. [1,2,3,4,5,6,7,8,9].
> > >
> > > ***Direction matrix as Householder reflection.*** An efficient method to generate an orthogonal matrix is the following: At iteration $k \in \mathbb{N}$, we generate the direction matrix $G_k$ as a single random Householder reflection i.e.
> > > \begin{equation}
> > > G_k := I - 2 v_k v_k^\intercal
> > > \end{equation}
> > > where $I$ is the identity $\mathbb{R}^{d \times d}$ and $v_k$ is a vector uniformly sampled from the sphere (i.e $v_k \in \mathbb{S}^{d- 1}$). Note that the cost of this method consists of two parts:
> > > - Generation of $v_k$: which consists in generating a Gaussian vector and in normalizing it.
> > > - The outer product $v_k v_k^\intercal$ for which modern implementations exploit parallelization/vectorization.
> > >
> > > The identity matrix can be generated and stored offline. Note that since it is very sparse, it can be stored using a sparse format (e.g. COO[10]). In this way, we can save resources in high-dimensional settings. The computation of the gradient approximation $g_k$ at iteration $k \in \mathbb{N}$ is
> > > \begin{equation*}
> > > 	g_k(x_k) := \frac{d}{\ell} \sum\limits_{i = 1}^\ell \frac{f(x_k + h_k G_k e_i) - f(x_k - h_k G_k e_i) }{2h_k}G_k e_i.
> > > \end{equation*}
> > > Thus, it "uses" only the first $\ell$ columns of $G_k$. Therefore, considering $\ell$ constant (as we proposed), we can store offline a (truncated) identity $I_{d, \ell}$ and compute the outer product truncating $v_k^\intercal$.  We report the time cost of generating a set of directions using this procedure and random (Gaussian and Spherical) directions in the case $\ell = d$ (i.e. the most time-expensive setting). Mean and standard deviation are indicated using 500 repetitions.
> > >
> > > ***
> > > d | Random Gaussian                 | Random Spherical | Householder
> > > ***
> > > 2 | 9.27e-7 $\pm$ 7.96e-7 | 5.49e-6 $\pm$ 2.05e-6 | 9.32e-6 $\pm$ 3.43e-6
> > >
> > > 4 | 1.30e-6 $\pm$ 7.21e-7 | 6.56e-6 $\pm$ 2.63e-6 | 1.12e-5 $\pm$ 5.79e-6
> > >
> > > 8 | 2.18e-6 $\pm$ 6.06e-7 | 8.01e-6 $\pm$ 5.32e-6 | 1.11e-5 $\pm$ 5.20e-6
> > >
> > > 16 | 5.69e-6 $\pm$ 1.61e-6 | 1.15e-5 $\pm$ 4.10e-6 | 1.18e-5 $\pm$ 7.20e-6
> > >
> > > 32 | 1.78e-5 $\pm$ 6.42e-6 | 2.49e-5 $\pm$ 1.33e-5 | 1.16e-5 $\pm$ 7.25e-6
> > >
> > > 64 | 6.58e-5 $\pm$ 7.03e-6 | 7.74e-5 $\pm$ 1.95e-5 | 1.62e-5 $\pm$ 3.79e-6
> > >
> > > 128 | 2.73e-4 $\pm$ 2.37e-5 | 2.98e-4 $\pm$ 2.45e-5 | 3.32e-5 $\pm$ 4.02e-6
> > >
> > > 256 | 1.26e-3 $\pm$ 2.79e-5 | 1.36e-3 $\pm$ 2.90e-5 | 1.20e-4 $\pm$ 1.04e-4
> > >
> > > 512 | 5.50e-3 $\pm$ 1.63e-4 | 5.91e-3 $\pm$ 1.22e-4 | 1.22e-3 $\pm$ 3.82e-4
> > >
> > > 1024 | 2.16e-2 $\pm$ 6.92e-4 | 2.41e-2 $\pm$ 7.35e-4 | 4.83e-3 $\pm$ 2.26e-3
> > >
> > > 2048 | 8.92e-2 $\pm$ 8.19e-2 | 1.04e-1 $\pm$ 1.03e-1 | 2.40e-2 $\pm$ 3.87e-2
> > > ***
> > >
> > >
> > >
> > > The resources of the machine used to perform this experiment are described in Appendix C. Note that, our procedure is more expensive than random directions only in small dimensional settings (i.e. for $d \leq 16$). However, for higher dimensional cases, it scales better in time (i.e. it is cheaper than random directions). Moreover, note that the highest cost in this procedure is the outer product which can be efficiently computed in GPU (and thus the time cost can be reduced by exploiting it). We will extend Appendix D including this table and other details. Moreover, in order to complete the answer to your question (and the Reviewer's KbSU fourth question) we have to compute and compare the performance in function values using this algorithm instead of random directions. To do that, we repeated the numerical experiments plotting the computational time in the x-axis, and reported the results in the global response (see Figure 2) as requested by Reviewer kbSU (the choice of the parameters is described in Section 4 and Appendix C). As we can observe, orthogonal directions still provide better performance than random directions. Such results confirm the empirical observations of [11].
> > >
> > > **References**
> > > 1. A. Genz. Methods for generating random orthogonal matrices.
> > > 2. F. Mezzadri. How to generate random matrices from the classical compact groups.
> > > 3. K. Choromanski, M. Rowland, W. Chen, and A. Weller. Unifying orthogonal monte carlo methods.
> > > 4. A. Hedayat and W. D. Wallis. Hadamard matrices and their applications.
> > > 5. Å. Björck. Numerics of gram-schmidt orthogonalization.
> > > 6. T. W. Anderson, I. Olkin, and L. G. Underhill. Generation of random orthogonal matrices.
> > > 7. A. Barvinok. Approximating orthogonal matrices by permutation matrices.
> > > 8. C. Rusu and L. Rosasco. Fast approximation of orthogonal matrices and application to pca.
> > > 9. C. Boutsidis and A. Gittens. Improved matrix algorithms via the subsampled randomized hadamard transform.
> > > 10. P. Virtanen et. al. SciPy 1.0: Fundamental Algorithms for Scientific Computing in Python.
> > > 11. A. Berahas, L. Cao, K. Choromanski and K. Scheinberg. A Theoretical and Empirical Comparison of Gradient Approximations in Derivative-Free Optimization.

---

### Official Review · Reviewer_KbSU · 2023-07-04

**Soundness:** 3 good
**Presentation:** 3 good
**Contribution:** 3 good
**Rating:** 6
**Confidence:** 4

**Summary:**

This paper proposed a structured zeroth-order estimator for non-smooth optimization. The proposed algorithm using this estimator could achieve optimal convergence rate for non-smooth convex optimization, also achieve a convergence rate in terms of Goldstein stationarity for non-smooth non-convex optimization. Numerical experiments are provided to show the efficiency of the proposed algorithm.

**Strengths:**

The theory is well-rounded with both theoretical and numerical evidences. The proposed structured zeroth-order estimator is the first one for non-smooth optimization. The use of the Goldstein stationary is pretty novel and an interesting direction to further explore.

**Weaknesses:**

(Please respond to the Questions section directly) The design of the zeroth-order estimator may cost more time, due to the fact that it requires sampling orthogonal matrices; The problem studied doesn’t cover stochastic situations; The numerical experiments are not adequate.

**Questions:**

Major:
1. Could the authors talk about possible applications of the proposed method? For example one could argue that zeroth-order smooth optimization can be applied to black-box attack of neural nets. But I’m not aware of the application of non-smooth situation.

2. The design of the zeroth-order estimator may cost more time, due to the fact that it requires sampling orthogonal matrices (either using QR or other matrix operations). More specifically, I’m wondering how the proposed method behaves comparing to simply sample $\ell$ vectors (with/without replacement) from the canonical orthogonal basis.

3. Regarding the numerical experiments. The choice of $\ell$ seem to be arbitrary. One could imagine that with a larger $\ell$, the per-iteration convergence would be faster, but it would cost more time to construct the estimator. Can the author give some insight on how to choose $\ell$ in an efficient manner? Another question related is that would the convergence and numerical behavior behave better if we vary $\ell$ in each iteration?

4. For the numerical experiment. I understand that due to page limit the authors moved a lot of details into the appendix, but it should be necessary to at least include the problem on which the experiments are conducted. Also since all the plots in the numerical experiments use “function evaluation” as ax-axis. I’m wondering what would the plot look like if we could the CPU time, since the proposed method could consume more time to construct the orthogonal matrix.

5. This is a personal comment and the authors may consider this for future works. The proposed method and analysis are all for deterministic optimization, whereas modern machine learning is about stochastic optimization. It would be interesting to see the convergence behavior of the proposed method for stochastic optimization problems, where the smooth stochastic case has been studied[1]. For zeroth-order non-smooth stochastic optimization, Goldstein stationarity seems to be necessary.


Minor:
1. Please add some reference in the abstract, e.g. line 7 “Recently,… improve performance”.

References:
[1] Balasubramanian, Krishnakumar, and Saeed Ghadimi. "Zeroth-order nonconvex stochastic optimization: Handling constraints, high dimensionality, and saddle points." Foundations of Computational Mathematics (2022): 1-42.

**Limitations:**

The limitation is well stated in weakness and question sections.

I’m not aware of any potential negative social impact of this work.

---

> ### Author Rebuttal · Authors · 2023-08-07
>
> We thank the reviewer for his/her comments and his/her suggestions. We answer here to his/her questions
>
> ## Questions
> 1. An example of an application of our algorithm we are working on is gain tuning in robotics.
> 2. Also coordinate directions (i.e. random canonical bases) provide good results. However, if the target function is "sparse" in the sense that the intrisic dimensionaly is smaller than the dimension (e.g. only some dimensions are relevant), sampling random canonical bases can provide no improvement while with random rotation we should be able to cover such settings.
> 3. In practice, the choice of $\ell$ depends on the function evaluation budget i.e. the maximum number of function evaluations that we can spend to face the optimization problem. Such budget depends on the time-cost of the single function evaluation and on the amount of time we want to spend to solve such a problem. According to the theory, it is better to perform less iterations with large $\ell$ instead of many iteration with small $\ell$. Thus a possible criterion can be to choose $\ell$ as large as possible (according to time constraints and computational resources). We thank the reviewer for the interesting idea on varying the number of directions $\ell$ per iteration. To the best of our knowledge there is no finite-difference method that analyze such a setting and it can be an interesting research direction.
> 4. We agree with the reviewer that it could be interesting to plot the CPU time vs function values. We repeated the experiments and we will include such plots either in Appendix or in the Numerical Experiments section. We included such plots in the global answer (i.e. the pdf file) - see Figure 2. In that case the orthogonal matrix is generated as a single Householder reflector (see Appendix D), in that way, we can precompute the truncated identity $I \in \mathbb{R}^{d \times \ell}$ and at every iteration we just have to generate a vector $v \in \mathbb{S}^{d - 1}$ and compute the outer product. The parameters are chosen as indicated in the paper (see Appendix C). Note that in order to make the comparison fair, the number of iterations performed by methods with multiple directions is smaller than the number of iterations performed by single direction methods (more precisely, given a budget of $T = 1000$ function evaluations, the number of iterations performed is $T/(2\ell)$). According to these experiments, we can still observe an advantage in using orthogonal directions. Moreover, we want to underline that in the literature there are different (and faster) methods to generate orthogonal matrices (see references in Appendix D). We want also to underline that in order to understand this kind of properties, we should perform an exhaustive empirical analysis which is out-of-the-scope of this work. However, note that previous works confirm that structured methods provide better practical performance (see e.g. [1]).
> 5. We thank the reviewer for the suggestion and we confirm that a research direction that we are considering consists in extending these results to the stochastic setting (considering different noise models). In this work, we wanted to introduce the first analysis for non-smooth structured zeroth and in particular the Smoothing Lemma for structured directions providing the basis to analysis structured zeroth order methods in non-smooth setting.
>
> **Minor:** we will add some references.
>
> **Reference**
> 1. A. S. Berahas, L. Cao, K. Choromanski, and K. Scheinberg. A theoretical and empirical comparison of gradient approximations in derivative-free optimization. Foundations of Computational Mathematics, 22(2):507–560, Apr 2022

---

> > ### Comment · Reviewer_KbSU · 2023-08-12
> > **Rebuttal acknowledgement**
> >
> > Thank you for your detailed responses to my comments and questions. I believe that paper is useful contribution to the field and would like to keep my score.

---

### Official Review · Reviewer_VJ1k · 2023-07-07

**Soundness:** 3 good
**Presentation:** 3 good
**Contribution:** 2 fair
**Rating:** 6
**Confidence:** 3

**Summary:**

Zeroth-order optimization is the sub-field of optimization concerned with solving
$$ x^{\star} \in \operatorname*{argmin}_{x \in \mathbb{R}^d}$$
_without_ using gradient information. This paper studies the most general case, where $f$ is assumed neither smooth nor convex. They propose a structured approach to the standard finite-difference gradient approximation, which means that the sampling directions are taken to be orthogonal.

The main contributions of this paper are theoretical; it provides a fine-grained analysis of the convergence rate of the proposed algorithm under a variety of assumptions. The theoretical claims are supported by two numerical experiments.

**Strengths:**

- The first 4 pages are easy to read, and position the problem under consideration (orthogonal sample directions for non-smooth optimization) nicely with respect to prior work
- This paper is definitely a technical advancement on prior work.
- The mathematical technique displayed in this paper is impressive. As an example, I enjoyed the use of the Goldstein subdifferential. The correct notion of "approximate stationarity" in this setting is quite subtle but you handle it very well.
 - In Corollaries 1, 2, and 4 I appreciated that you provided convergence guarantees in both the constant step-size and decreasing step-size settings. Both are useful


**Weaknesses:**

- Overall I found this paper to be a little too heavy on theory and a little too light on experiment. I would suggest relegating some of the parameter configurations of Corollaries 2, 4, and 5 to an appendix and adding an additional experiment (see next point).
 - I'd strongly suggest adding more experiments; in my opinion the two simple functions you tested do not provide enough data to draw any conclusions. It would be particularly interesting to see more non-smooth experiments.

Minor points:
 - In line 235 "the ball centered in 0" should be "the ball centered at x"
 - In line 239 you write $\partial f_h(x_I)$ but I think you mean $\partial_h f(x_I)$.
 - Why is there a staircase pattern in the purple graph in Figure 1?

**Questions:**

1. In the related work section, you state that your method achieves the optimal dependence on dimension and then cite [1]. But this paper deals with stochastic zeroth-order optimization with a particular "two-point" query model, whereas in your paper you consider the non-stochastic/noise-free setting.

I did look at [1] but could not find a clear statement of the optimal dependence on dimension in the noise-free setting (I believe it is $\mathcal{O}(d)$). Could you make this a bit clearer in your exposition? Also, what is the authoritative reference for the optimal $d$-dependence for noise-free DFO?

[1] J. C. Duchi, M. I. Jordan, M. J. Wainwright, and A. Wibisono. Optimal rates for zero-order convex optimization: The power of two function evaluations.

**Limitations:**

Yes.

---

> ### Author Rebuttal · Authors · 2023-08-07
>
> We thank the reviewer for his/her observations and comments. We answer here to his/her questions
>
> ## Weakness
> The main goal of this article is to provide the first analysis of a structured zeroth-order algorithm in non-smooth setting providing the mathematical tools (i.e. the Smoothing Lemma) required to analyze its convergence, and hopefully future extensions (e.g. stochastic setting). The purpose of the numerical experiments section is to empirically show the properties indicated by the theory.
>
> An exhaustive empirical comparison is out of the scope of this work but we are considering it as a research direction.
>
> We performed other experiments that we will include in the paper (some of them are included in the global answer - see Figure 1). However, we want to underline that in order to understand the practical behavior of the algorithm, an exhaustive empirical analysis should be performed keeping in account also the other parameters (e.g. stepsize choice or the discretization choice) and this is out of the scope of this work.
>
>
> **Minor:** in figure 1 we analyze the impact of the number of directions $\ell$. In particular, we plotted in x-axis the number of function evaluations. Now, since to compute the gradient estimator we need to perform $2\ell$ function evaluations, we repeated $2\ell$ times the target function values (this is explained in line 323-324).
>
> ## Questions
> The reference is the same, note that by replacing the stochastic oracle with a noise-free oracle the proofs (of the propositions) follow the same line.

---

> > ### Comment · Reviewer_VJ1k · 2023-08-16
> > **Thanks**
> >
> > Thanks to the authors for their response! I have no further questions.

---

### Official Review · Reviewer_eK7Y · 2023-07-08

**Soundness:** 3 good
**Presentation:** 3 good
**Contribution:** 3 good
**Rating:** 6
**Confidence:** 2

**Summary:**

This paper introduces and analyzes  a structured finite difference algorithm for non-smooth optimization problems. The algorithm is built on a smooth approximation of the nonsmooth loss function and a structured finite difference approximation. The convergence of the proposed algorithm in non-smooth convex, non-smooth non-convex, smooth convex and smooth non-convex cases are studied.

**Strengths:**

1. A simple algorithm using structured finite difference approximation of the gradient is proposed.

2. The convergence behavior of the proposed algorithm in four cases are studied.

**Weaknesses:**

I think the relation between the convergence of the proposed method and the number of directions chosen is not clearly analyzed. The error in Theorem 1 is only linear in \ell. While numerical experiments suggests \ell has an impact on the convergence rate.

**Questions:**

1. In Theorem 1 and Corollary 1, the error is linear in \ell. The convergence rate seems only depends on \theta. Does \ell affect the convergence rate?

2. From the theories in this paper, setting \theta towards 1/2 provides better convergence rate. But what is the hurt by doing that?

**Limitations:**

The limitation is not discussed.

---

> ### Author Rebuttal · Authors · 2023-08-07
>
> We thank the reviewer for his/her useful comments. We answer here to his/her questions.
>
> ## Questions
> **Dependence on $\ell$ in the rate:** Yes, the number of directions $\ell$ affects the convergence rate but only in the constants. Indeed,  Theorem $1$ states that
>
> \begin{equation}
> \\mathbb{E}[f(\bar{x}_k) - \min f] \leq S_k / A_k = \frac{1}{\sum\_{i=0}^k \alpha_i } \Big( \frac{\|\| x_0 - x^*\| \|^2}{2} + c\frac{d L\_0^2}{\ell} \sum\limits\_{i=0}^k \alpha_i^2 + L\_0 \sum\limits\_{i = 0}^k \alpha_i h_i \Big).
> \end{equation}
>
> Note that the impact of the dimension in the rates depends on the choice of the stepsize $\alpha_k$ and the discretization parameter $h_k$. Since the second term on the right part of the inequality depends on $\frac{d}{\ell}$, in order to reduce the impact of the dimension in the rate and obtain the optimal complexity, we need to include a $\sqrt{\frac{\ell}{d}}$ in the stepsize $\alpha_k$. The same observation holds for Corollary $1$ (specifically in points $(i)$ and $(ii)$). We will include these observations in the discussion paragraph below Corollary $1$.
>
> **Choice of $\theta$ in the stepsize:** As for classic methods like subgradient descent and stochastic gradient descent, the choice of $\theta$ towards $1/2$ is the best choice we can provide. Indeed, note that the second term in the right part of the inequality of Theorem $1$ (see the inequality above) is an error term which does not depend on the discretization sequence $h_k$ and thus it is bounded if and only if $\alpha_k^2$ goes to $0$. Thus, ideally, we would choose a sequence that decreases as fast as possible. However, if $\alpha_k$ goes to $0$ too fast, we might have no convergence, for this reason we need to take $\alpha_k \not\in \ell^{1}$ and $\alpha_k^2 \in \ell^{1}$. The result is not surprising and is in line with the classical results obtained by studying algorithms with errors or based on subgradients.
>
> ## Limitations
> Limitations of the algorithm are discussed in Appendix E. We will add a reference in the main paper.

---

### Author Rebuttal · Authors · 2023-08-07

Several reviewers have requested more experiments, including nonsmooth functions, and comparison in terms of CPU time. We include some of these in the attached pdf. We run these experiments $20$ times and provide the mean and standard deviation of the results.

---

### Decision · Program_Chairs · 2023-09-21

**Decision:**

Accept (poster)

**Comment:**

The paper achieves the optimal complexity in the non-smooth convex case and provided convergence rates for other settings. The zeroth-order estimator is novel in the context of non-smooth optimization. Some reviewers have requested more experiments, which the authors have provided during the rebuttal. I encourage the authors include these new experiments in the appendix of the updated version.